# Homeostatic Adaptation of Optimal Population Codes under Metabolic Stress

**Yi-Chun Hung[1], Gregory W. Schwartz[2], Emily A. Cooper[3,4], Emma Alexander[1]**

[1]Department of Computer Science, Northwestern University
[2]Departments of Ophthalmology and Neuroscience, Feinberg School of Medicine, Northwestern University
[3]Herbert Wertheim School of Optometry & Vision Science, University of California, Berkeley
[4]Helen Wills Neuroscience Institute, University of California, Berkeley
ychung@u.northwestern.edu    emilycooper@berkeley.edu
{greg.schwartz,ealexander}@northwestern.edu

## Abstract

Information processing in neural populations is inherently constrained by metabolic resource limits and noise properties. Recent data, for example, shows that neurons in mouse visual cortex can go into a "low power mode" in which they maintain firing rate homeostasis while expending less energy. This adaptation leads to increased neuronal noise and tuning curve flattening in response to metabolic stress. These dynamics are not described by existing mathematical models of optimal neural codes. We have developed a theoretical population coding framework that captures this behavior using two surprisingly simple constraints: an approximation of firing rate homeostasis and an energy limit tied to noise levels via biophysical simulation. A key feature of our contribution is an energy budget model directly connecting adenosine triphosphate (ATP) use in cells to a fully explainable mathematical framework that generalizes existing optimal population codes. Specifically, our simulation provides an energy-dependent dispersed Poisson noise model, based on the assumption that the cell will follow an optimal decay path to produce the least-noisy spike rate that is possible at a given cellular energy budget. Each state along this optimal path is associated with properties (resting potential and leak conductance) which can be measured in electrophysiology experiments and have been shown to change under prolonged caloric deprivation. We analytically derive the optimal coding strategy for neurons under varying energy budgets and coding goals, and show that our method uniquely captures how populations of tuning curves adapt while maintaining homeostasis, as has been observed empirically.

## 1 Introduction

Animals have limited access to calories, which constrains their brains' ability to expend energy to represent information. Additionally, energy budgets are inconsistent over the lifetime of an organism, which must adapt in times of scarcity while maintaining as much functionality as possible. Padamsey et al. have recently shown that under long-term calorie deprivation, mice lose the ability to discriminate fine-grained visual details while keeping coarser information, and that they do so with systematic changes in biophysical cell properties that result in a flattening of neuronal tuning curves (Padamsey et al., 2022). Perhaps surprisingly, this energy-saving strategy does not result in a significant change in spike counts, demonstrating the importance of firing rate homeostasis even when energy budgets are tight.

This energy-saving strategy motivates us to develop an analytical and simulation-grounded framework that describes how metabolically-stressed neurons can optimally change to produce noisier, cheaper spike trains rather than reducing their firing rates, consistent with Padamsey et al.'s empirical observations. Firing rate homeostasis under metabolic stress, along with direct but tractable accounting for ATP use, is the key to correctly modeling how real tuning curves change. The standard analytical approaches of limiting spike rates (Ganguli & Simoncelli, 2010; Wang et al., 2016a)

can predict optimal populations under a single metabolic state, but fails to capture how codes adapt to changing conditions.

The efficient coding hypothesis (Barlow et al., 1961) asserts that populations of sensory neurons are optimized to maximize information about the environment subject to resource constraints. While the impact of coding objectives (Ganguli & Simoncelli, 2010; 2014; Manning et al., 2024; Wang et al., 2016a) and noise properties (Laughlin et al., 1998; Manning et al., 2024; Ecker et al., 2011; Wang et al., 2016a) on optimal population codes have been studied in depth, the effect of energy constraints and in particular their variation over time remains underexplored. Simple models of energy constraints (e.g., on mean firing rate in Ganguli & Simoncelli (2010); Yerxa et al. (2020); Rast & Drugowitsch (2020) or maximum firing rate in Wang et al. (2016a;b); Laughlin (1981)) are analytically convenient, but they fail to capture the complex trade-offs determining energetic costs in real cells, such as subthreshold activity and biophysical adaptations (Padamsey & Rochefort, 2023). We will show that these simplified constraints lead to inaccurate predictions of neurons' adaptation to metabolic stress (i.e., reduction of energy budget). Specifically, our model predicts tuning curve flattening closely in line with Padamsey et al. (2022), while previous population coding models capture either the shortening (Ganguli & Simoncelli, 2010) or widening (Wang et al., 2016a) behavior of energy-limited tuning curves, but not both.

We capture this behavior with two novel constraints: an energy budget (linked directly to ATP consumption, firing rate noise, and easily-measured cell properties) and an approximation of homeostasis. Remarkably, our biophysically-motivated constraints do not complicate the analysis of coding strategies; instead, they provide a simple extension of existing models to characterize the dynamics of neural coding strategies under metabolic stress. The contributions of our framework are as follows:

- We introduce two mathematically-tractable constraints that are the first to accurately describe real neurons' recently-characterized response to metabolic stress.
- Our framework generalizes previous models, and can recover disparate results from the literature with simplified parameters, offering a united mathematical explanation for the partial successes of previous work.
- Biophysical simulation grounds key parameters of our model, including a dispersed Poisson noise model that realistically describes signal degradation in calorie-deprived neurons.

## 2 RELATED WORK

**Optimal neural codes.** Since the foundational work by Barlow et al. (1961), the efficient coding hypothesis has been extensively evaluated across a variety of contexts, leading to the development of numerous information-theoretic models with strong empirical support from both neural and perceptual data (Ganguli & Simoncelli, 2016; Manning et al., 2024; Vacher & Mamassian, 2023; Risau-Gusman, 2020; Zhaoping et al., 2011; Laughlin, 1981). A variety of coding goals have been analyzed, from entropy maximization (Laughlin, 1981) to mutual information maximization (Brunel & Nadal, 1998; Wei & Stocker, 2016; Wang et al., 2016b; Manning et al., 2024; Ganguli & Simoncelli, 2010; 2014) to $L_p$ norm error minimization (Wang et al., 2016a; Ganguli & Simoncelli, 2010; 2014; Manning et al., 2024).

Optimal coding models are often built on Fisher information (FI) (Wang et al., 2016a; Ganguli & Simoncelli, 2010; 2014; Manning et al., 2024; Wang et al., 2016b), which describes the local sensitivity of a population response around a stimulus value. One reason for this is that FI can be estimated more easily from measurements of tuning curves than entropy-based information metrics. Another is that FI optimization is closely related to standard coding objectives: maximizing $\log$ FI approximates mutual information maximization (Brunel & Nadal, 1998; Wei & Stocker, 2016), while maximizing power functions of FI minimizes $L_p$ loss in the asymptotic limit (Wang et al., 2012). Coding goals of particular interest here are the "infomax" (mutual information maximization, through $\log$ FI maximization) and "discrimax" (discriminability maximization, through $-FI^{-1}$ maximization, implying $L_2$ error minimization by the Cramér Rao bound) objectives.

Optimal neuronal populations can be predicted directly from FI-based objectives, provided that significant simplifying assumptions are imposed on tuning curve shapes. For example, Wei & Stocker (2015) assumes homogeneous tuning widths or amplitudes across the population. Usefully, Ganguli

& Simoncelli (2010) introduced a tiling assumption that enables a more compact representation of population codes through two continuous functions: gain and density, denoted by $g(\cdot)$ and $d(\cdot)$, respectively. These functions are then used to modulate a base shape (e.g., Gaussian or sigmoid) to construct heterogeneous population codes. Wang et al. (2012) also adopt the tiling assumption, with a fixed gain function.

**Constrained optimization.** Constraints shape optimal codes just as strongly as coding objectives. Constraints in prior literature can typically be categorized into three types: energy, coding capacity, and population size. Energy-based constraints, such as those based on mean firing rate (Ganguli & Simoncelli, 2010; Yerxa et al., 2020; Rast & Drugowitsch, 2020) or maximum firing rate (Wang et al., 2016a;b; Laughlin, 1981), reflect the energetic costs of neuronal firing. Coding capacity, expressed as the sum of the square root of FI, has been employed in Wei & Stocker (2016); Wang et al. (2012) and generalized to other power laws in Morais & Pillow (2018). The population size constraint Ganguli & Simoncelli (2010); Yerxa et al. (2020) reflects the finite number of neurons available during neural processing.

The joint influence of coding objectives and constraints on shaping optimal population codes has also been explored in recent work. Morais & Pillow (2018) show how direct trade-offs between coding objectives and FI-based constraints can obscure the relationship between these factors. Rast & Drugowitsch (2020) describe an adaptation procedure to jointly determine objectives and constraints from neural data, but their model does not allow changes in co-firing patterns like those caused by tuning curve widening. Most similar to our work, Wang et al. (2016b) analyzes ON-OFF population codes under metabolic stress by modeling both energy costs and noise levels as power laws of firing rate. However, their energy cost model is inconsistent with Padamsey et al. (2022) and they do not connect the two terms through a noise-energy trade-off. None of these constraints connect directly to ATP use, due to the biophysical complexity of real neurons.

**Energy costs beyond spike counts and homeostatic mechanisms.** ATP is the energy currency of cells, yet most models of energy consumption in neurons use the simplifying assumption that spikes have a fixed cost and dominate the neuron's ATP usage. In fact, spikes are not typically the largest part of a neuron's ATP budget, costing only 22% of energy use (Harris et al., 2012). Reversal of ion fluxes from postsynaptic receptors (50%) and reversal of leak sodium entry via the $Na^+/K^+$ exchanger (20%) together dominate ATP usage in neurons (Harris et al., 2012). We use a biophysical model of a neuron that estimates ATP usage from all three of these sources, creating a more realistic energy constraint than spike rate alone.

In fact, one of the key features of our neuron simulation is that it estimates different energy rates while maintaining a fixed spike rate. This result is consistent with Padamsey et al. (2022) and relies on systematic changes in cell parameters observed in metabolically stressed mouse cortex. Homeostatic maintenance of a fixed firing rate is a commonly observed feature in neural circuits, often involving the balance of excitation and inhibition (Liang et al., 2024).

It is worth noting that the approximating a neuronal energy cost through firing rate is commonly adopted in recent literature (Tavoni, 2025; Koren & Panzeri, 2022; Koren et al., 2025; Gutierrez & Denève, 2019) for analyzing changes in intrinsic cell-level and/or network level properties. Although these works use firing rate as an energy proxy same as in Ganguli & Simoncelli (2010); Yerxa et al. (2020); Rast & Drugowitsch (2020); Wang et al. (2016a;b); Laughlin (1981), they do not consider the effect of metabolic stress on neurons, as reported in Padamsey et al. (2022). In fact, these models cannot reiterate a central finding of this recent work: that spike rates remain the same as energy use is reduced.

**Recently-discovered neural mechanism.** Padamsey et al. (2022) show that neurons change their biophysical state so that they can maintain their mean firing rates while expending less energy. Primarily, they modify electrical properties to reduce the current associated with receiving input spikes, because reversing this current costs significant ATP through ion pumps. Essentially, by sitting closer to the spiking threshold and receiving a smaller signal from each incoming spike, the cell uses less energy for signal-carrying spikes but also becomes readier to fire in response to stochastic inputs and system noise, which makes them more vulnerable to false firings. Because these false firings are strictly positive, they raise the zero values at the edges of their tuning curves (widening), which requires a compensatory drop in peak values (flattening) to maintain homeostasis.

## 3 ANALYTICAL MODEL

We consider a population of $N$ neurons jointly encoding a scalar stimulus $s$ with their mean firing rate determined by tuning curves $\{h_n(s)\}_{n=1}^N$. Assuming conditionally independent firing rates, the FI of the population can be computed from the tuning curves as

$$FI(s; E) = \sum_{n=1}^n \frac{h_n'(s)^2}{\text{Var}(h_n(s); E)}. \tag{1}$$

Based on the neuron simulation described in Sec. 4, energy-varying noise can be expressed as a dispersed Poisson distribution, with a dispersion factor $\eta_\kappa(E)$ that captures variability in neural responses under different activity levels $\kappa$ and energy budgets $E$:

$$\text{Var}(h_n(s); E) = \eta_\kappa(E)h_n(s). \tag{2}$$

An optimal population, under an energy budget $E$, maximizes the expectation with respect to probability $p(s)$ of a target function $f$ on the FI, while maintaining firing rate homeostasis (expected firing rate $R_n$) within each neuron:

$$\underset{h_1(\cdot),\ldots,h_N(\cdot)}{\text{argmax}} \int p(s)f\left(FI(s; E)\right)\text{d}s, \quad \text{s.t.} \int p(s)h_n(s)\,\text{d}s = R_n, \ \forall n. \tag{3}$$

Notably, if $f(x) := \log x$, the formulation corresponds to infomax; whereas if $f(x) := -x^{-1}$, it corresponds to discrimax.

### 3.1 APPROXIMATION WITH TILING

Solving eq. (3) is challenging without parametrizing the tuning curves $h_n(s)$, due to the vast space of arbitrary continuous functions. To mitigate this difficulty, we parameterize the tuning curves using two continuous functions — gain $g(s)$ and density $d(s)$—following the approach of Ganguli & Simoncelli (2010). Specifically, the tuning curve is defined as $h_n(s) = g(s)\hat{h}(D(s) - D(s_n))$, where $s_n$ denotes the preferred stimulus of the $n$-th neuron. Here, $D(s)$ represents the cumulative function of the density $d(s)$, given by $D(s) = \int_{-\infty}^s d(s)\,ds$. We assume that $p(s)$ varies more smoothly than any tuning curve $h_n(s)$ and analyze the case in which the base shape $\hat{h}(\cdot)$ is Gaussian.

To approximate the summation in eq. (3) in terms of $g(s)$ and $d(s)$, we adopt the same tiling property as in Ganguli & Simoncelli (2010); Wang et al. (2012). Under this assumption, the Fisher information is approximated as:

$$FI(s; E) \approx \eta_\kappa(E)^{-1}g(s)d(s)^2. \tag{4}$$

We can approximate the homeostasis constraint, and generalize it to the population level, as follows:

$$p(s)g(s) = R(s)d(s). \tag{5}$$

This *approximate homeostasis constraint* can be visualized as a rectangular (height $g(s_n)$, width $1/d(s_n)$) approximation of each Gaussian tuning curve and a locally constant $p(s)$. Detailed derivations of eqs. 4 and 5 are provided in Secs. A to C.

Combining the approximations in eq. (4) and eq. (5) gives the following population optimization:

$$\underset{g(\cdot),d(\cdot)}{\text{argmax}} \int p(s)f\left(\eta_\kappa(E)^{-1}g(s)d(s)^2\right)\text{d}s, \quad \text{s.t.} \quad p(s)g(s) = R(s)d(s). \tag{6}$$

It is evident that this optimization problem is only bounded when an additional constraint is imposed on $g(s)$ and/or $d(s)$. We propose the following generalized energy constraint:

$$\int p(s)g(s)^\alpha \text{d}s = E, \quad \text{where} \ \alpha \geq 1. \tag{7}$$

This model is consistent with Padamsey et al. (2022), which shows that energy savings come from reducing synaptic conductance (analogous to gain) rather than other cell parameters that widen the tuning curve (analogous to density), which we interpret as compensation to maintain homeostasis. We will relate the energy budget $E$ to physical energy expenditure and fix the value of $\alpha$ for real neurons in Sec. 4.

## 3.2 ANALYTICAL SOLUTION

We analytically solve the proposed optimization framework for various objective functions (see Tab. 1) by eliminating $d(s)$ via the approximate homeostasis constraint and applying the Lagrangian method. Here, we outline the key steps in the derivation for the infomax case, with other objective functions and full details included in Sec. D. To solve the optimization problem defined by eqs. (6) and (7), we eliminate $d(s)$ by substituting from the approximate homeostasis constraint in eq. (5), yielding the following formulation:

$$\max_{g(\cdot)} \int p(s) \log \left( \eta_\kappa(E)^{-1} g(s)^3 p(s)^2 R(s)^{-2} \right) \mathrm{d}s, \quad \text{s.t.} \quad \int p(s) g(s)^\alpha \, \mathrm{d}s = E. \qquad (8)$$

The corresponding Lagrangian with multiplier $\lambda$ is given by:

$$\mathcal{L}(g, \lambda) = \int p(s) \log \left( \eta_\kappa(E)^{-1} g(s)^3 p(s)^2 R(s)^{-2} \right) \mathrm{d}s - \lambda \left( \int p(s) g(s)^\alpha \, \mathrm{d}s - E \right). \qquad (9)$$

Setting $\partial_g \mathcal{L} = 0$ and using eq. (7) provides the optimal gain, with density following from eq. (5):

$$g(s) = E^{1/\alpha}, \qquad (10)$$
$$d(s) = E^{1/\alpha} p(s) / R(s). \qquad (11)$$

The resulting FI is obtained with eq. (4) and the discriminability bound from the Cramér Rao bound, summarized in Tab. 1, along with the analytical solutions corresponding to $L_p$ error objective functions. It is interesting to point out that our model predicts that both density (i.e., $1/\text{width}$) and gain scale on the order of $E^{1/\alpha}$, while the Fisher information scales on the order of $\eta_\kappa(E)^{-1} E^{3/\alpha}$. Additionally, we show in Sec. N that the analytical solutions in eqs. (10) and (11) are robust to differential correlations, which are the dominant violation of the independent noise assumption in real neurons (Moreno-Bote et al., 2014). This robustness extends to significant changes in noise correlations across metabolic states.

Table 1: **Analytical solutions in our framework.** For comparison to other methods, see Sec. K.

|  | infomax | discrimax | $L_p$ **error,** $p = -2\beta$ |
|---|---|---|---|
|  | $f(x) = \log x$ | $f(x) = -x^{-1}$ | $f(x) = -x^\beta, \ \beta < \frac{\alpha}{3}$ |
| **Density** $d(s)$ | $E^{\frac{1}{\alpha}} R(s)^{-1} p(s)$ | $\propto E^{\frac{1}{\alpha}} R(s)^{\frac{-\alpha-1}{\alpha+3}} p(s)^{\frac{\alpha+1}{\alpha+3}}$ | $\propto E^{\frac{1}{\alpha}} R(s)^{\frac{\alpha-\beta}{3\beta-\alpha}} p(s)^{\frac{\beta-\alpha}{3\beta-\alpha}}$ |
| **Gain** $g(s)$ | $E^{\frac{1}{\alpha}}$ | $\propto E^{\frac{1}{\alpha}} R(s)^{\frac{2}{\alpha+3}} p(s)^{\frac{-2}{\alpha+3}}$ | $\propto E^{\frac{1}{\alpha}} R(s)^{\frac{2\beta}{3\beta-\alpha}} p(s)^{\frac{-2\beta}{3\beta-\alpha}}$ |
| **Fisher info** $FI(s)$ | $\propto \frac{E^{\frac{3}{\alpha}} p(s)^2}{\eta_\kappa(E) R(s)^2}$ | $\propto \frac{E^{\frac{3}{\alpha}} p(s)^{\frac{2\alpha}{\alpha+3}}}{\eta_\kappa(E) R(s)^{\frac{2\alpha}{\alpha+3}}}$ | $\propto \frac{E^{\frac{3}{\alpha}} p(s)^{\frac{2\alpha}{3\beta-\alpha}}}{\eta_\kappa(E) R(s)^{\frac{2\alpha}{\alpha-3\beta}}}$ |
| **Disc. bound** | $\propto E^{\frac{-3}{2\alpha}} p(s)^{-1}$ | $\propto E^{\frac{-3}{2\alpha}} p(s)^{\frac{-\alpha}{\alpha+3}}$ | $\propto E^{\frac{-3}{2\alpha}} p(s)^{\frac{\alpha}{3\beta-\alpha}}$ |

## 3.3 RELATION TO EXISTING MODELS

The approximate homeostasis and energy constraints introduced in eqs. (5) and (7) are not only grounded in biophysiological evidence but also generalize commonly used constraints in existing models (see Tab. 2). Here, we first demonstrate how our constraints relate to those in Ganguli & Simoncelli (2010), followed by a discussion of the connection to Wang et al. (2012). It is evident that our energy constraint reduces to the mean firing rate (FR) constraint used in Ganguli & Simoncelli (2010) when setting $\alpha = 1$ and redefining $E$ as the mean firing rate denoted as $R$. The population size constraint can also be recovered from our constraints as follows:

$$E \stackrel{\alpha := 1}{=} \int p(s) g(s) \, \mathrm{d}s \stackrel{\text{approx. homeo.}}{=} \int R(s) d(s) \, \mathrm{d}s \stackrel{R(s) := \frac{E}{N}}{=} \frac{E}{N} \int d(s) \, \mathrm{d}s \Rightarrow \int d(s) \, \mathrm{d}s = N. \tag{12}$$

By jointly enforcing the conditions that yield the mean FR and population size constraints, we obtain $R(s) = R/N$, indicating that the population mean firing rate $R$ is evenly distributed across neurons.

It is important to note that Ganguli & Simoncelli (2010) uses the parameter $R$ to denote the total expected firing rate of a neuron population. The parameter $R_n$ in eq. (3) represents the expected firing rate of the $n^{\text{th}}$ neuron in the population. Meanwhile, $R(s)$ can be interpreted as the expected

firing rate of a neuron whose preferred stimulus is $s$, and can be regarded as a "continuous" version of $R_n$.

The derivation in eq. (12) leads us to observe that approximate homeostasis, though not an explicit requirement of Ganguli & Simoncelli (2010), emerges as a signature of optimality in their model (see Sec. E for proof of this proposition).

To reduce our constraint to the constraint of total FI states in Wang et al. (2012), we now set $\alpha = 3/2$:

$$E \overset{\alpha := \frac{3}{2}}{=} \int p(s)g(s)^{\frac{3}{2}} \, ds \overset{\text{approx. homeo.}}{=} \int R(s) \underbrace{\left( d(s)g(s)^{\frac{1}{2}} \right)}_{\sqrt{\text{FI}}} \, ds$$

$$\overset{R(s) := \frac{E}{C}}{=} \frac{E}{C} \int \sqrt{d(s)^2 g(s)} \, ds \quad \Rightarrow \quad \int \sqrt{d(s)^2 g(s)} \, ds = C, \tag{13}$$

where $C \in \mathbb{R}$ is the FI coding capacity. Under the assumption $\alpha = 3/2$, the above reduction implies that our energy constraint can be interpreted as a weighted version of a coding resource constraint, where the weighting function corresponds to the firing rate $R(s)$. The generalization of our model to the existing frameworks is summarized and illustrated in Fig. 1.

In summary, we show that our general case can be reduced to previous models in the literature. Specifically, by setting $\alpha = 1$ and using firing rate as an energy proxy reduces our constraint to that of Ganguli & Simoncelli (2010). Furthermore, when $\alpha = 3/2$ and the FI coding capacity is incorporated, our energy constraint reduces to the coding capacity constraint proposed in Wang et al. (2012). In the following section, we will describe the biophysical parameters that reduce the general case to a true ATP-based energy constraint.

Table 2: **Comparison of different works on neuron population codes.** Key parameters include the total number of neurons ($N$), mean firing rate ($R$, as a constant or a function of stimulus $s$), Fisher information coding capacity ($C$), energy budget ($E$).

| Works | Constraints | Constraints under Tiling | Noise | Low Energy |
|---|---|---|---|---|
| Ganguli & Simoncelli (2010) | mean FR population size | $\int p(s)g(s)ds = R$ $\int d(s)ds = N$ | Poisson | $R \downarrow$, shortens ($N$ fixed) |
| Wang et al. (2016a) | max FR total FI states | $g(s) \propto 1$ $\int \sqrt{g(s)d(s)^2}ds = C$ | Poisson or Gaussian | $C \downarrow$, widens ($g$ fixed) |
| Ours | energy approx. homeostasis | $\int p(s)g(s)^{\alpha}ds = E$ $\frac{g(s)}{d(s)} = \frac{R(s)}{p(s)}$ | Energy-dependent Dispersed Poisson | $E \downarrow$, flattens ($R(s)$ fixed) |

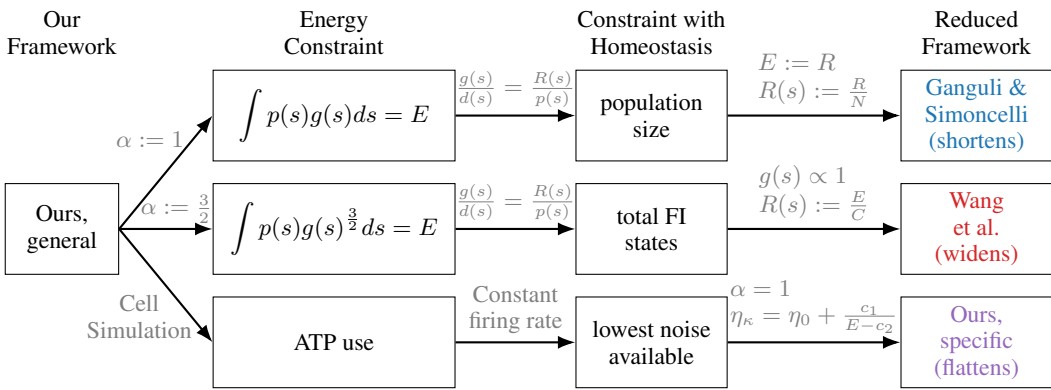

Figure 1: **A general solution.** Our analytical framework generalizes previous work (Ganguli & Simoncelli, 2010; Wang et al., 2016a) and predicts tuning curve flattening with a biophysical grounding in neuron simulation.

## 4 BIOPHYSICAL SIMULATION

We employ the biophysical simulator NEURON (Hines et al., 2022) to investigate the trade-offs between noise and energy. We simulate a single-compartment neuron based on a stochastic Hodgkin-Huxley model in response to an input spike, under a variety of settings. Specifically, we vary the three cellular parameters identified by Padamsey et al. (2022) as changing under metabolic stress: resting potential $v_{rest}$, leak channel conductance $g_{leak}$, and synaptic conductance $g_{syn}$. See Fig. 2 for an overview of the simulation pipeline, Sec. F for an intuitive explanation of each parameter's effect as well as a full list of simulation parameters, Sec. G for details of energy accounting, and the supplement for full simulation code.

Value ranges for $v_{rest}$ and $g_{leak}$ follow those of Padamsey et al. (2022), and we vary $g_{syn}$ from 0 to 250 $\mu S/cm^2$. This range allows us to exclude cell parameter triplets that lead to mean spike count over .8 or under .2 total during the 2 second simulation duration in the deterministic setting, so that stochastic simulation captures meaningful variance. From $10,000$ simulated trials for each condition, we evaluate the mean and variance of the spike count, denoted as $\mu_F$ and $\sigma_F^2$, as well as the energy expenditures associated with maintaining the resting state and signal activity (combining the cost of receiving and generating spikes), denoted as $\epsilon_{bg}$ and $\epsilon_{sig}$, respectively.

We then derive higher level statistics from the single-spike characteristics. First, we compute the energy cost $\epsilon_{total}$ associated with each cell state. This cost combines rest state and signaling energy, weighted by an activity level $\kappa$ (see Sec. H.1). This factor is key for modeling visual neurons, which have orders-of-magnitude changes in spike rates along the visual pathway (Goris et al., 2014). Then, we use the variance of spike count to characterize noise dispersion, details in the following section. Finally, we follow Padamsey et al. (2022) to measure tuning curve widening $w$ by asserting a Gaussian tuning curve via rescaled synaptic conductance $g_{syn}$, details in Sec. H.2. Note that this procedure associates a unique value of $g_{syn}$ to each $(v_{rest}, g_{leak})$ pair, reducing the degrees of freedom in our cell parameter space.

### 4.1 NOISE-ENERGY OPTIMALITY

Using the noise $\sigma_F^2$ and energy $\epsilon_{total}$ associated with each cell state $(v_{rest}, g_{leak}, \kappa)$, we assert two hypotheses in turn to predict biophysical and information theoretic changes under metabolic stress, see Fig. 3. First, we assert a strict homeostasis constraint to extract cell parameters corresponding to constant mean spike count (white plane in Fig. 3a) and extract the corresponding energy costs (Fig. 3b, energy costs across the extracted plane shown in d). We note in Fig. 3c that firing count variance saturates and then drops as $g_{syn}$ increases, as the probability of firing approaches 1. The variance is quadratic in mean spike count, consistent with a Bernoulli distribution, but scaled by a factor that grows under increased metabolic stress. Specifically, we fit the scale of the firing variance parabola for the dispersion $\eta_\kappa$:

$$\sigma_F^2 = \eta_\kappa \, \mu_F (1 - \mu_F). \tag{14}$$

Fig. 3e illustrates this fitting for a single $(v_{rest}, g_{leak})$ state. Although we fit the $\eta_\kappa$ based on mean and variance of spike count under single spikes, it can be generalized to spike trains (see Sec. H.3).

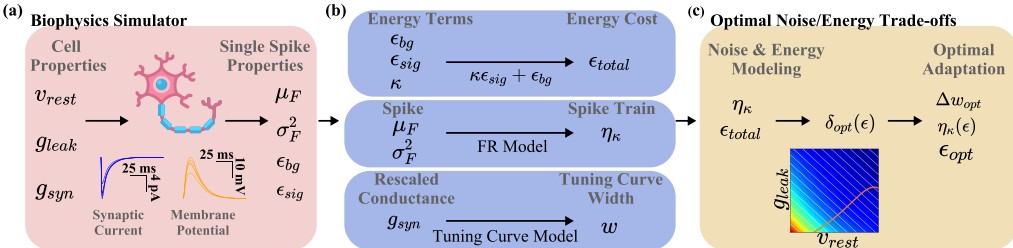

Figure 2: **Biophysical simulation of noise-optimal energy reduction.** (a) We change cell properties of a simulated neuron to determine their impact on cell firing and energy use. (b) Single spike properties are extended to tuning curve properties under varying cell states. (c) We define optimal paths as those that minimize firing rate variance at each energy consumption level, predicting a specific noise/energy trade-off that we incorporate into our Fisher Information optimization framework.

We then assert our second assumption: that the cell will adapt so that for every energy budget, noise will be minimized. Fig. 3f shows a heatmap of $\eta_\kappa$ associated with each cell state, with white lines indicating level sets of energy cost. Along each constant-energy line, the lowest noise point lies on the orange curve, which describes an optimal decay path through the cell parameter space. Specifically, this path is the solution of the following constrained optimization problem:

$$\delta_{opt}(\epsilon;\kappa) = \underset{v_{rest},\, g_{leak}}{\operatorname{argmin}}\; \eta_\kappa(v_{rest}, g_{leak}) \quad \text{s.t.} \quad \epsilon_{total}(v_{rest}, g_{leak};\kappa) = \epsilon. \tag{15}$$

To numerically solve this minimization problem, we fit $\eta_\kappa$ using a 17-parameter model and intersect it with a linear fit of $\epsilon_{total}$ (see Sec. H.4 for fitting details). Each $(v_{rest}, g_{leak})$ cell state along the optimal path $\delta_{opt}(\epsilon, \kappa)$ corresponds to both an energy cost and a noise level, providing the empirical trade-off between these terms, as well as a tuning curve width. Note that the shape of this curve varies with activity level $\kappa$, as shown in Fig. 3g. As $v_{rest}$ becomes less negative, signaling becomes cheaper but maintaining resting potential is more expensive, so only active cells deviate significantly from -75 mV. Additionally, changing the balance of background vs. signaling energy terms changes the slope of constant-energy-contours, which intersects with the nonlinear noise landscape to generate paths of different curvature (see Sec. I).

This simulation fixes the unknown terms $\alpha$ and $\eta_\kappa$ from our mathematical model as follows. First, we note that our optimal densities in Tab. 1 are all proportional to $E^{\frac{1}{\alpha}}$.

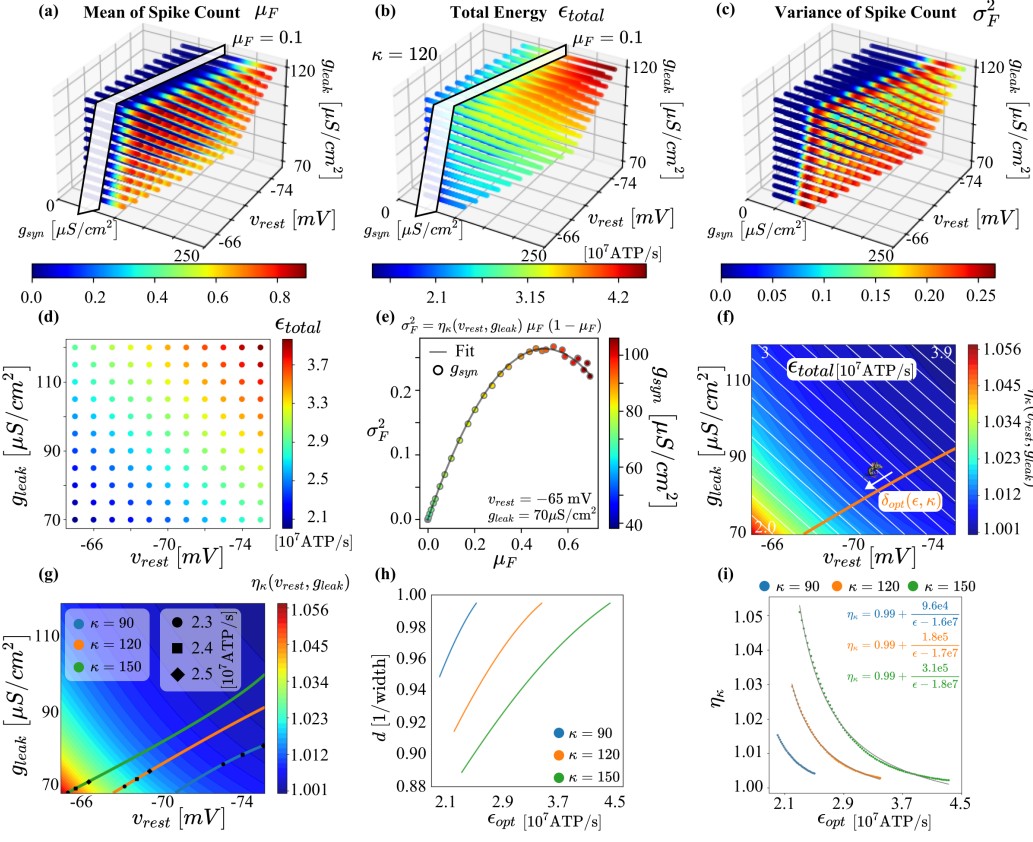

Figure 3: **Simulation results of optimal adaptations under varying signal activity levels and intermediate outcomes.** (a–c) Simulated results of total energy, and the mean and variance of spike count, respectively. (d) The energy corresponding to a fixed spike count, illustrated by the intersection in (a). (e) Fitted relationship between the mean and variance of spike count for an example pair of $v_{rest}$ and $g_{leak}$, used to define the dispersion $\eta_\kappa(v_{rest}, g_{leak})$. (f) Optimal adaptations obtained numerically by minimizing dispersion along each energy contour using the fitted energy and dispersion. (g) Optimal adaptations shown across different levels of signal activity $\kappa$. (h, i) The resulting density and dispersion as a function of optimal energy, respectively.

When we define $E$ in units of ATP/s as

$$E(\kappa) = a_1(\kappa)\epsilon_{opt} + a_2(\kappa), \tag{16}$$

we get the optimal density scales as $d(s) \propto E^{\frac{1}{\alpha}} = (a_1(\kappa)\epsilon_{opt} + a_2(\kappa))^{\frac{1}{\alpha}}$. Since Fig. 3h shows that, across activity levels, the optimal density has an affine relationship with $\epsilon_{opt}$, it follows immediately that

$$\alpha = 1. \tag{17}$$

See Sec. J for fits of $\vec{a}(\kappa)$ values from simulation, as well as $\vec{b}(\kappa)$, $\vec{c}(\kappa)$, and $\eta_0(\kappa)$ defined below.

The noise dispersion along the optimal adaptations shown in Fig. 3i follows the intuitive trend that a decrease in energy leads to an increase in noise. Higher activity and tighter energy constraints lead to larger changes in noise levels. In general, the noise dispersion takes the form

$$\eta_\kappa(\epsilon_{opt}) = \eta_0(\kappa) + \frac{b_1(\kappa)}{\epsilon_{opt} - b_2(\kappa)} = \eta_0(\kappa) + \frac{c_1(\kappa)}{E(\kappa) - c_2(\kappa)}, \tag{18}$$

showing a direct trade-off between noise and energy in both physical units and the analytical energy budget $E$. Returning to our optimized solutions in Tab. 1, we note that noise only indirectly affects the optimal density and gain, by shaping the optimal path and therefore setting $\alpha$. It also directly scales the Fisher information by a factor of $\eta_\kappa(E)^{-1}$, allowing empirically-derived $\eta_\kappa$ values to function as a plug-and-play term when estimating Fisher information for subsequent analyses, such as the evaluation of perceptual thresholds.

## 5 CONSISTENCY WITH DATA

Here, we will show that our method accurately predicts the tuning curve flattening effect seen in mouse cortex (Padamsey et al., 2022), while existing models incorrectly predict shortening or widening. We assume an infomax-optimal Gaussian tuning curve for a uniformly-distributed stimulus, and explore predicted changes under metabolic stress (see Sec. L for details). Mouse data in Padamsey et al. (2022), illustrated in Fig. 4a, shows a 32% widening of the tuning curve and a statistically insignificant change in firing rate, which we model as an exact match in spike count. In Fig. 4b, our model reiterates the 32% widening and 29% reduction in ATP consumption reported in Padamsey et al. (2022). To accomplish both simultaneously, we fit $a_1$ and $a_2$ from eq. (16) to satisfy $a_2/(a_1\epsilon_{ctr}) = 0.19625$, where $\epsilon_{ctr}$ denotes the energy expenditure in the control condition. In the infomax case with a uniform prior, we achieve a perfect tuning curve match because our approximation of homeostasis is exact. See Sec. L for homeostatic errors in other settings.

We compare our method to existing, non-homeostatic frameworks (Ganguli & Simoncelli, 2010; Wang et al., 2016a). In Fig. 4c, Ganguli & Simoncelli (2010) constrains the mean firing rate directly, with a fixed tuning curve width derived from the (fixed) population size. Because the tuning curve cannot match the widening observed in data, we reduce the firing rate to match the tuning curve peak instead. This requires a direct 24% reduction in the firing rate, violating homeostasis by -24% under our uniform prior assumption. In Fig. 4d, Wang et al. (2016a) maintains a constant max firing rate while reducing coding capacity, with a 24% reduction in coding capacity required to widen by 32%, resulting in a homeostatic violation of 32%. See Sec. M for a discussion of model likelihoods.

## 6 DISCUSSION

Changes in energy availability are a fact of life for all sensing organisms, and a key factor in evolutionary fitness. Indeed, there is good evidence that neural circuits maintain spike rate homeostasis across a range of contexts, perhaps as an essential part of keeping activity in neural circuits balanced (Hengen et al., 2013; Wu et al., 2020).

We have characterized the noise-energy trade-off in a simulation that connects a tractable and generalizing mathematical framework directly to ATP use. It also makes specific, testable predictions about how basic biophysical properties of cells should change under metabolic stress, factoring in activity level changes across the visual pathway. Specifically, the optimal paths in Fig. 3g predict specific relationships between resting potential and leak conductance for neurons of different spike

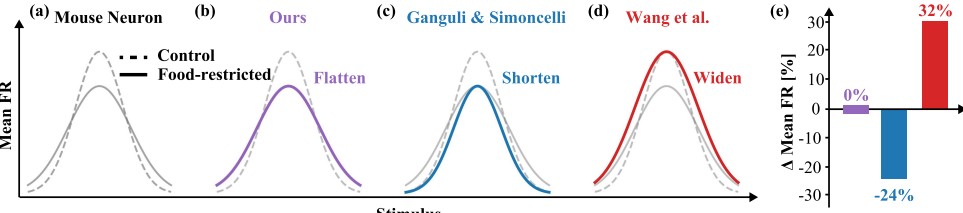

Figure 4: **Comparison of optimal tuning curves in control versus food-restricted mouse neocortex (L2/3).** (a-d) The figure illustrates how an example tuning curve in an optimal population adapts to a tightening of energy-related constraints. Under metabolic stress, real tuning curves flatten (a, based on Padamsey et al. (2022)). Our model (b, purple) predicts this flattening, while existing models predict either shortening (c, blue Ganguli & Simoncelli (2010)) or widening (d, red Wang et al. (2016a)). (e) We compute the change in mean firing rate for each method under a uniform prior. Only our model exhibits firing rate homeostasis.

rates as they face metabolic pressure. These properties are readily measured in patch-clamp experiments, and future work will perform these experiments in matched types of neurons from control and food-restricted animals.

In summary, we analytically derive and predict that both density (i.e., $1/\text{width}$) and gain scale on the order of $E^{1/\alpha}$, while the Fisher information scales on the order of $\eta_\kappa(E)^{-1} E^{3/\alpha}$ (see Sec. 3). Additionally, we demonstrate that our model encompasses the competing models (Ganguli & Simoncelli, 2010; Wang et al., 2016a) as special cases. As a result, it preserves their explanatory power and, in particular, matches the ability of Ganguli & Simoncelli (2016) to produce signatures of optimality that have been found in brain areas for five measured attributes (three visual and two auditory). In Sec. 4, we use a stochastic Hodgkin–Huxley-like model to calibrate the parameters, yielding $\alpha = 1$ and $\eta_\kappa(E) = \eta_0 + \frac{c_1}{E-c_2}$. Finally, in Sec. 5, we show that the analytical solution with the calibrated $\alpha$ and $\eta_\kappa$ accurately predicts the tuning-curve flattening effect reported in Padamsey et al. (2022).

Future work can address current limits of our model. Mathematically, we seek to provide more general tractable approximations of homeostasis conditions, particularly for non-Gaussian and non-tiling populations. Sigmoids and Gabors are tuning curve families of particular interest. Our single-compartment biophysical neuron model can also be extended to a more complicated simulation to capture higher-fidelity cell behaviors. It is also possible that cell recordings will show that, contrary to our central assumption, metabolic stress responses are driven primarily by non-coding factors and are not information-theoretically optimal, calling for additional modelling.

Tractable, grounded models of metabolic impacts on neural codes are needed in order to ask new kinds of optimization questions, interpret existing data, and incorporate accelerating progress in our understanding of metabolism. The new optimization questions we can ask include: how can population codes be optimized to handle a *range* of energy budgets? Does an animal's lifetime or early life access to calories impact their neural codes? What can we tell about the needs and priorities of an animal from how their codes change under metabolic stress? In short, we can consider optimality in terms of long-term survival strategies rather than at snapshot states. The answer to these questions may lead to reinterpretations of existing data that has been used as evidence of particular optimization functions, as in Manning et al. (2024); Ganguli & Simoncelli (2016). And while Padamsey et al. (2022) describes changes that occur on the timescale of weeks, it is possible that some metabolic stress responses occur more quickly, with implications for the interpretation of neural data from food-deprived animals, common in behavioral literature. Though our model is based heavily on data from Padamsey et al. (2022), a rapidly emerging understanding of the importance of metabolic and homeostatic factors in visual system function and nervous system health (Walls et al., 2025; Lin et al., 2025; Etchegaray & Ravichandran, 2011; Meng et al., 2025; Kim et al., 2025; Sian-Hulsmann et al., 2024) suggests that more data in this area may be available soon.

## ACKNOWLEDGMENTS

This work was supported by Taiwan-Northwestern Doctoral Scholarship. We would like to acknowledge Mark Agrios and the NSF-Simons National Institute for Theory and Mathematics in Biology (NITMB) for valuable discussions.

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

## A APPROXIMATE FISHER INFORMATION

We assume following

- Convolution tiling property: $\sum_n \frac{\hat{h}'(D(s)-D(s_n))^2}{\hat{h}(D(s)-D(s_n))} \approx I_{\text{conv}}$.

- $g(s)$ is much smoother than the function $\hat{h}(s)$.

- Each neuron responds independently.

The first assumption is commonly adopted in the previous literature (Ganguli & Simoncelli, 2010; Wang et al., 2016a) and the second one is implicitly used in Ganguli & Simoncelli (2010). Based on these assumptions, we can approximate the total Fisher information $FI(s)$ as:

$$
\begin{aligned}
FI(s; E) &= \sum_n \frac{h_n'(s)^2}{\eta_\kappa(E)h_n(s)} \quad \text{(If dispersed Poisson and by independence)} \\
&\approx \eta_\kappa(E)^{-1} \sum_n \frac{\left(g(s)\hat{h}'(D(s) - D(s_n))d(s)\right)^2}{g(s)\hat{h}(D(s) - D(s_n))} \\
&= \eta_\kappa(E)^{-1} \sum_n \frac{g(s)d(s)^2\hat{h}'(D(s) - D(s_n))^2}{\hat{h}(D(s) - D(s_n))} \\
&\approx \eta_\kappa(E)^{-1} g(s)d(s)^2 I_{\text{conv}}.
\end{aligned}
\tag{19}
$$

The approximation relies on the assumptions. In our analysis, we further assume that the constant $I_{\text{conv}}$ is equal to 1, as it does not influence the analytical solution of the proposed framework. For the self-consistency, we also derive the first equality in Sec. B.

## B DISPERSED POISSON NOISE MODELING

Assuming that the firing rate of the $n$-th neuron, denoted $r_n$, is sampled from a dispersed Poisson distribution approximated by $\mathcal{N}(h_n(s), \eta_\kappa(E)h_n(s))$, we can derive the Fisher information (FI) as

follows:

$$FI_n(s; E) := \mathbb{E}\left[\left(\frac{\partial \log p(r_n; s)}{\partial s}\right)^2\right] \tag{20}$$

$$= (h'_n(s))^2 \, \mathbb{E}\left[\left(\frac{-1}{2h_n(s)} + \frac{r - h_n(s)}{\eta_\kappa(E)h_n(s)} + \frac{(r - h_n(s))^2}{2\eta_\kappa(E)h_n(s)^2}\right)^2\right] \tag{21}$$

$$= (h'_n(s))^2 \left(\frac{1}{4h_n(s)^2} + \frac{1}{\eta_\kappa(E)h_n(s)} + \frac{3}{4h_n(s)^2} - \frac{1}{2h_n(s)^2}\right) \tag{22}$$

$$= \frac{h'_n(s)^2}{\eta_\kappa(E)h_n(s)} + \frac{h'_n(s)^2}{2h_n(s)^2} \tag{23}$$

$$\approx \frac{h'_n(s)^2}{\eta_\kappa(E)h_n(s)} \tag{24}$$

where the second line follows from the definition of FI, and the third line is obtained by applying moment properties of the Gaussian distribution. The last approximation is based on the non-negligible firing rates (e.g., $h_n(s) \gg \frac{\eta_\kappa(E)}{2}$). Note that the second term in eq. (23) is due to the Gaussian approximation of the dispersed Poisson. It can be more elegantly addressed by introducing the quasi-likelihood function McCullagh (2019).

## C  APPROXIMATE HOMEOSTASIS

**Proposition C.1** (Approximate Homeostasis). *Assume the following:*

- *The probability density $p(s)$ and the gain $g(s)$ of the stimulus $s$ are much smoother than $\hat{h}(D(s) - D(s_n))$, where $D(s)$ is the cumulative function of the density $d(s)$.*

- *The functions $\hat{h}(\cdot)$ and $D(s)$ are differentiable.*

*Then, the following approximation holds:*

$$\int p(s)h_n(s)\,ds = \int p(s)g(s)\hat{h}(D(s) - D(s_n))\,ds \approx p(s_n)\frac{g(s_n)}{d(s_n)}\int \hat{h}(x)\,dx. \tag{25}$$

In the proof, we first exploit the smoothness of the probability density $p(s)$, followed by a first-order Taylor approximation of $D(s) - D(s_n)$ around $s_n$, and finally apply the change of variable $x = d(s_n)(s - s_n)$.

*Proof.*

$$\begin{aligned}
\int p(s)h_n(s)\,ds &= \int p(s)g(s)\hat{h}\left(D(s) - D(s_n)\right)\,ds \\
&\approx p(s_n)g(s_n)\int \hat{h}\left(D(s) - D(s_n)\right)\,ds \\
&\approx p(s_n)g(s_n)\int \hat{h}\left(d(s_n)(s - s_n)\right)\,ds \\
&= p(s_n)\frac{g(s_n)}{d(s_n)}\int \hat{h}(x)\,dx,
\end{aligned} \tag{26}$$

where the first approximation uses the smoothness assumption of $p(s)$, and the second approximation applies a first-order Taylor expansion of $D(s)$ around $s_n$. $\square$

Followed by the approximation in theorem C.1, we further relax the approximation to the population level and assume the normalized $\hat{h}(x)$. Then, we obtain the approximate homeostasis constraint:

$$p(s)\frac{g(s)}{d(s)} = R(s). \tag{27}$$

# D  DERIVATION OF ANALYTICAL SOLUTIONS

## D.1  INFOMAX

$$\underset{g(\cdot),d(\cdot)}{\operatorname{argmax}} \int p(s) \log \left( \eta_\kappa(E)^{-1} g(s) d(s)^2 \right) \mathrm{d}s$$

$$\text{s.t.} \quad p(s) \frac{g(s)}{d(s)} = R(s) \tag{28}$$

$$\int p(s) g(s)^\alpha \mathrm{d}s = E, \quad \text{where} \ \alpha \geq 1.$$

By eliminating $d$, we have

$$\underset{g(\cdot)}{\operatorname{argmax}} \int p(s) \log \left( \eta_\kappa(E)^{-1} g(s)^3 p(s)^2 R(s)^{-2} \right) \mathrm{d}s, \quad \text{s.t.} \quad \int p(s) g(s)^\alpha \mathrm{d}s = E. \tag{29}$$

The corresponding Lagrangian with multiplier $\lambda$ is given by:

$$\mathcal{L}(g,\lambda) = \int p(s) \log \left( \eta_\kappa(E)^{-1} g(s)^3 p(s)^2 R(s)^{-2} \right) \mathrm{d}s - \lambda \left( \int p(s) g(s)^\alpha \mathrm{d}s - E \right). \tag{30}$$

Setting $\partial_g \mathcal{L} = 0$, we obtain

$$\frac{\partial \mathcal{L}}{\partial g} = \frac{3p(s)}{g(s)} - \lambda \alpha p(s) g(s)^{\alpha-1} = 0 \tag{31}$$

$$\Rightarrow \quad g(s) = \left( \frac{3}{\lambda\alpha} \right)^{\frac{1}{\alpha}} \ (\text{Assume} \ p(s) > 0). \tag{32}$$

Then using the energy constraint, we have

$$E = \int p(s) g(s)^\alpha \mathrm{d}s = \int p(s) \frac{3}{\lambda\alpha} \mathrm{d}s = \frac{3}{\lambda\alpha}, \tag{33}$$

Therefore, we get the optimal solution:

$$g(s) = E^{1/\alpha}, \tag{34}$$

$$d(s) = E^{1/\alpha} R(s)^{-1} p(s). \tag{35}$$

## D.2  DISCRIMAX

$$\underset{g(\cdot),d(\cdot)}{\operatorname{argmax}} \int -p(s) \left( \eta_\kappa(E)^{-1} g(s) d(s)^2 \right)^{-1} \mathrm{d}s$$

$$\text{s.t.} \quad p(s) \frac{g(s)}{d(s)} = R(s) \tag{36}$$

$$\int p(s) g(s)^\alpha \mathrm{d}s = E, \quad \text{where} \ \alpha \geq 1.$$

By eliminating $d$, we have

$$\underset{g(\cdot)}{\operatorname{argmax}} \int -p(s) \left( \eta_\kappa(E)^{-1} g(s)^3 p(s)^2 R(s)^{-2} \right)^{-1} \mathrm{d}s, \quad \text{s.t.} \quad \int p(s) g(s)^\alpha \mathrm{d}s = E. \tag{37}$$

The corresponding Lagrangian with multiplier $\lambda$ is given by:

$$\mathcal{L}(g,\lambda) = \int -p(s) \left( \eta_\kappa(E) g(s)^{-3} p(s)^{-2} R(s)^2 \right) \mathrm{d}s - \lambda \left( \int p(s) g(s)^\alpha \mathrm{d}s - E \right). \tag{38}$$

Setting $\partial_g \mathcal{L} = 0$, we obtain

$$\frac{\partial \mathcal{L}}{\partial g} = 3p(s)^{-1} \eta_\kappa(E) g(s)^{-4} R(s)^2 - \lambda \alpha p(s) g(s)^{\alpha-1} = 0 \tag{39}$$

$$\Rightarrow \quad g(s) = \left( \frac{3\eta_\kappa(E) R(s)^2}{\lambda\alpha p(s)^2} \right)^{\frac{1}{\alpha+3}} \tag{40}$$

Then using the energy constraint, we have

$$E = \int p(s)g(s)^\alpha \mathrm{d}s = \int p(s) \left( \frac{3\eta_\kappa(E)R(s)^2}{\lambda\alpha p(s)^2} \right)^{\frac{\alpha}{\alpha+3}} \mathrm{d}s \tag{41}$$

$$= \left( \frac{3\eta_\kappa(E)}{\lambda\alpha} \right)^{\frac{\alpha}{\alpha+3}} \underbrace{\left( \int p(s)^{1-\frac{2\alpha}{\alpha+3}} R(s)^{\frac{2\alpha}{\alpha+3}} \mathrm{d}s \right)}_{A_{disc}} \tag{42}$$

$$\Rightarrow \quad \lambda = \frac{3\eta_\kappa(E)}{\alpha} \left( \frac{A_{disc}}{E} \right)^{\frac{\alpha+3}{\alpha}}, \tag{43}$$

where we denote the integral in eq. (42) as a scalar $A_{disc}$. Note that $A_{disc}$ is constant with respect to $s$ once the integral has been performed, but will vary if $p(s)$, $R(s)$, or $\alpha$ changes. Because this term will not change the shape of any predicted distributions (e.g., gain, density, or Fisher information), we suppress these dependencies in our notation fo convenience.

Plugging $\lambda$ to $g(s)$, we have

$$g(s) = \left( \frac{3\eta_\kappa(E)R(s)^2}{\lambda\alpha p(s)^2} \right)^{\frac{1}{\alpha+3}} = \left( \frac{3\eta_\kappa(E)R(s)^2}{\frac{3\eta_\kappa(E)}{\alpha} \left( \frac{A_{disc}}{E} \right)^{\frac{\alpha+3}{\alpha}} \alpha p(s)^2} \right)^{\frac{1}{\alpha+3}} \tag{44}$$

$$= \left( \frac{R(s)^2}{\left( \frac{A_{disc}}{E} \right)^{\frac{\alpha+3}{\alpha}} p(s)^2} \right)^{\frac{1}{\alpha+3}} \tag{45}$$

$$= \left( \frac{E^{\frac{1}{\alpha}} R(s)^{\frac{2}{\alpha+3}}}{A_{disc}^{\frac{1}{\alpha}} p(s)^{\frac{2}{\alpha+3}}} \right) \propto E^{\frac{1}{\alpha}} R(s)^{\frac{2}{\alpha+3}} p(s)^{\frac{-2}{\alpha+3}} \tag{46}$$

Therefore, we get the optimal solution:

$$g(s) \propto E^{\frac{1}{\alpha}} R(s)^{\frac{2}{\alpha+3}} p(s)^{\frac{-2}{\alpha+3}}, \tag{47}$$

$$d(s) \propto E^{\frac{1}{\alpha}} R(s)^{\frac{-\alpha-1}{\alpha+3}} p(s)^{\frac{\alpha+1}{\alpha+3}}. \tag{48}$$

### D.3 GENERAL

$$\operatorname*{argmax}_{g(\cdot),d(\cdot)} \int -p(s) \left( \eta_\kappa(E)^{-1} g(s)d(s)^2 \right)^\beta \mathrm{d}s$$

$$\text{s.t.} \quad p(s)\frac{g(s)}{d(s)} = R(s) \tag{49}$$

$$\int p(s)g(s)^\alpha \mathrm{d}s = E, \quad \text{where } \alpha \geq 1.$$

By eliminating $d$, we have

$$\operatorname*{argmax}_{g(\cdot)} \int -p(s) \left( \eta_\kappa(E)^{-1} g(s)^3 p(s)^2 R(s)^{-2} \right)^\beta \mathrm{d}s, \quad \text{s.t.} \quad \int p(s)g(s)^\alpha \, \mathrm{d}s = E. \tag{50}$$

The corresponding Lagrangian with multiplier $\lambda$ is given by:

$$\mathcal{L}(g,\lambda) = \int -p(s) \left( \eta_\kappa(E)^{-\beta} g(s)^{3\beta} p(s)^{2\beta} R(s)^{-2\beta} \right) \, \mathrm{d}s - \lambda \left( \int p(s)g(s)^\alpha \, \mathrm{d}s - E \right). \tag{51}$$

Setting $\partial_g \mathcal{L} = 0$, we obtain

$$\frac{\partial \mathcal{L}}{\partial g} = -p(s) \cdot 3\beta\eta_\kappa(E)^{-\beta} g(s)^{3\beta-1} p(s)^{2\beta} R(s)^{-2\beta} - \lambda\alpha p(s)g(s)^{\alpha-1} = 0, \tag{52}$$

$$\Rightarrow \quad g(s) = \left( \frac{\lambda\alpha}{3\beta} \eta_\kappa(E)^\beta \cdot \frac{R(s)^{2\beta}}{p(s)^{2\beta}} \right)^{\frac{1}{3\beta-\alpha}}. \tag{53}$$

Then using the energy constraint, we have

$$E = \int p(s)g(s)^\alpha ds = \left(\frac{\lambda\alpha}{3\beta}\eta_\kappa(E)^\beta\right)^{\frac{\alpha}{3\beta-\alpha}} \underbrace{\int p(s)^{1-\frac{2\alpha\beta}{3\beta-\alpha}} R(s)^{\frac{2\alpha\beta}{3\beta-\alpha}} ds}_{A_{gen}}. \tag{54}$$

$$\Rightarrow \quad \lambda = \frac{3\beta}{\alpha}\eta_\kappa(E)^{-\beta}\left(\frac{E}{A_{gen}}\right)^{\frac{3\beta-\alpha}{\alpha}}. \tag{55}$$

Therefore, we get the optimal solution:

$$g(s) \propto E^{\frac{1}{\alpha}} R(s)^{\frac{2\beta}{3\beta-\alpha}} p(s)^{\frac{-2\beta}{3\beta-\alpha}}, \tag{56}$$

$$d(s) \propto E^{\frac{1}{\alpha}} R(s)^{\frac{\alpha-\beta}{3\beta-\alpha}} p(s)^{\frac{\beta-\alpha}{3\beta-\alpha}}. \tag{57}$$

# E    RELATION BETWEEN APPROXIMATE HOMEOSTASIS CONSTRAINT AND OPTIMALITY

**Proposition E.1** (Framework in Ganguli & Simoncelli (2010) Implies Approx. Homeostasis). *Let the optimization problem be formulated as*

$$\max_{g(\cdot),d(\cdot)} \int_s p(s) \cdot f\left(g(s)d(s)^2\right) ds,$$

$$s.t. \quad \int_s p(s)g(s)ds = R, \tag{58}$$

$$\int_s d(s)ds = N,$$

*where R and N are constants. Then any optimal solution $(g^*, d^*)$ must satisfy*

$$p(s)\frac{g(s)}{d(s)} = \frac{R}{N}. \tag{59}$$

*Proof.* We form the Lagrangian:

$$L(g, d, \lambda_1, \lambda_2) = \int_s p(s)f(g(s)d(s)^2)ds$$
$$- \lambda_1\left(\int_s p(s)g(s)ds - R\right) \tag{60}$$
$$- \lambda_2\left(\int_s d(s)ds - N\right).$$

Taking the first-order conditions, we have:

$$\frac{\partial L}{\partial g} = p(s)\frac{\partial f(x)}{\partial x}d(s)^2 - \lambda_1 p(s) = 0, \tag{61}$$

$$\frac{\partial L}{\partial d} = 2p(s)\frac{\partial f(x)}{\partial x}g(s)d(s) - \lambda_2 = 0. \tag{62}$$

Dividing the two equations:

$$\frac{\lambda_2}{\lambda_1} = \frac{2p(s)\frac{\partial f(x)}{\partial x}g(s)d(s)}{\frac{\partial f(x)}{\partial x}p(s)d(s)^2} = \frac{2p(s)g(s)}{d(s)}. \tag{63}$$

Since $\lambda_1, \lambda_2$ are constants (w.r.t. $s$), the ratio $\frac{2p(s)g(s)}{d(s)}$ must also be constant. Letting $C = \frac{\lambda_2}{2\lambda_1}$, we get

$$p(s)\frac{g(s)}{d(s)} = C. \tag{64}$$

To identify this constant, multiply both sides by $d(s)$ and integrate:

$$\int_s p(s)g(s)ds = C\int_s d(s)ds, \tag{65}$$

$$\Rightarrow C = \frac{R}{N}. \tag{66}$$

Thus, the optimal solution must satisfy

$$p(s)\frac{g(s)}{d(s)} = \frac{R}{N}. \tag{67}$$

$\square$

# F    SIMULATION OVERVIEW

Recall that current reversal is a significant energy cost, so that at a high level each of these parameters has the following effect:

**Resting potential** $v_{rest}$**:**    When $v_{rest}$ is less negative, receiving spikes is cheaper, because it takes less of a voltage change (and therefore less current) to reach the spiking threshold. Input current must be reversed by ion pumps using ATP. Perhaps counterintuitively, a less-negative $v_{rest}$ also makes sitting at rest more expensive. This is because a less negative $v_{rest}$ shifts the balance between the reversal potentials of $Na^+$ and $K^+$, which requires counteracting a higher influx of $Na^+$ and costs more energy to maintain (see Sec. G for details and relevant equations). This cost trade-off between maintaining rest potential and reversing the current associated with incoming spikes is why the optimal paths shown in Fig. 3g vary in shape as a function of the input activity level of the cell.

**Leak channel conductance** $g_{leak}$**:**    Leak channels are a major factor in the overall membrane conductance. Using $V = IR = I/g$ we see that this mechanism also reduces the total current needed to make a voltage change in the cell.

**Synaptic conductance** $g_{syn}$**:**    This prevents the cell from firing excessively by counteracting the changes in $v_{rest}$ and $g_{leak}$ that reduce the energy cost of reaching spiking threshold. It scales down the impact of each incoming spike so that the mean firing rate of the cell is maintained. Essentially, for a cell that is twice as ready to fire, we adjust the synaptic conductance so that each incoming spike only creates half the change it usually would. This rescaled state is not identical in function to the original because the smaller distance to threshold, particularly in combination with the higher voltage variability also observed in this state, results in more accidental output spikes. (We will illustrate this effect with a figure of simulated voltage traces, showing a cell that requires three sequential input spikes to fire: in the well-fed case, two large voltage steps will not cause an output spike even in the presence of noise, but in the starving case, two smaller steps result in a smaller gap from threshold, which can be overcome by noise and cause an unintended output spike.) Note that this variable is how we inject a specific tuning curve into the model, following Padamsey et al. (2022).

We additionally summarize how we configure the parameters in the simulation (see Tab. 3) and list the notation (see Tab. 4) in this paper.

Table 3: Simulation Parameters

| Category | Value |
|---|---|
| membrane capacitance | 1 $\mu$F/cm$^2$ |
| membrane leaky channel conductance (control) | 0.12 mS/cm$^2$ |
| membrane reversal potential for passive current (control) | $-75$mV |
| membrane action potential threshold | $-50$mV |
| ion channel model file | stch_carter_subtchan.ses |
| Na$^+$ reversal potential | 55 mV |
| K$^+$ reversal potential | $-90$ mV |
| Na$^+$ channel total conductance | 35 mS/cm$^2$ |
| single Na$^+$ channel conductance | 20 pS |
| Na$^+$ channel stochasticity | False |
| K$^+$ Delayed Rectifier total conductance | 4 mS/cm$^2$ |
| single K$^+$ Delayed Rectifier conductance | 20 pS |
| K$^+$ Delayed Rectifier stochasticity | False |
| K$^+$ subchan total conductance | 0.18 mS/cm$^2$ |
| single K$^+$ subchan conductance | 20 pS |
| K$^+$ subchan stochasticity | True |
| soma diameter | 8 $\mu$m |
| soma length | 8 $\mu$m |
| synapse input type | synaptic conductance |
| synapse rise time constant (double-exponential model) | 1 ms |
| synapse decay time constant (double-exponential model) | 10 ms |
| synapse reversal potential | 0 mV |
| synapse input noise model | $\mathcal{N}(g_{syn}, \frac{g_{syn}}{10})$ |
| stimulus spiking event | single spike |
| stimulus stochasticity | False |
| simulation initial potential | -75 mV |
| simulation duration | 2000 ms |
| simulation step | 0.1 ms |

Table 4: Notation in modeling and framework

| Name | Notation |
|---|---|
| resting potential | $v_{rest}$ |
| membrane leaky channel conductance | $g_{leak}$ |
| Maximum synapse conductance | $g_{syn}$ |
| mean of spike count | $\mu_F$ |
| variance of spike count | $\sigma_F^2$ |
| background energy consumption | $\epsilon_{bg}$ |
| signal energy consumption | $\epsilon_{sig}$ |
| mean of firing rate | $\mu_{FR}$ |
| variance of firing rate | $\sigma_{FR}^2$ |
| stimulus-dependent synaptic conductance (gaussian shape) | $G(s)$ |
| tuning curve width | $w$ |
| signal activity | $\kappa$ |
| total energy cost | $\epsilon_{total}$ |
| optimal adaptation | $\delta_{opt}(\epsilon)$ |
| noise dispersion along optimal adaptation | $\eta_\kappa(E)$ |
| stimulus | $s$ |
| preferred stimulus of the $n$-th neuron | $s_n$ |
| total Fisher information | $FI(s)$ |
| Fisher information of the $n$-th neuron | $FI_n(s)$ |
| objective function | $f(\cdot)$ |
| stimulus probability density function | $p(s)$ |
| gain | $g(s)$ |
| density | $d(s)$ |
| base shape (gaussian) | $\hat{h}(s)$ |
| tuning curve of the $n$-th neuron | $h_n(s)$ |
| firing rate of the $n$-th neuron | $R_n$ |
| firing rate of approximate homeostasis | $R(s)$ |
| energy cost of the energy constraint | $E$ |
| total number of neurons | $N$ |
| exponent in the energy constraint | $\alpha$ |
| exponent in the objective function of the general case | $\beta$ |

## G    Biophysical simulation

The overview of the simulation pipeline is shown in Fig. 2. We focus on three key cellular properties — resting potential $v_{rest}$, leak channel conductance $g_{leak}$, and synaptic conductance $g_{syn}$ — as the primary variables in our simulation, based on findings by Padamsey et al. (2022), who reported that neurons adapt their encoding approach through changes in these parameters.

We employ a single-compartment simulation model that includes three ionic channels, one passive (leaky) channel, and a double-exponential conductance-based synapse (see Tab. 3). Additionally, the ionic channel modeling files are based on Padamsey et al. (2022). A single input spiking event is delivered to the synapse at the beginning of each simulation trial. In the simulation, we configure a three-dimensional parameter space defined by $v_{rest}$, $g_{leak}$, and $g_{syn}$. Specifically, $v_{rest}$ and $g_{leak}$ are varied over the ranges of $-75$ to $-65$ mV and $0.07$ to $0.12$ mS/cm$^2$, respectively. For each pair of $v_{rest}$ and $g_{leak}$, we further tune $g_{syn}$ to ensure that the resulting mean spike count remains within a physiologically valid range based on the $g_{syn}$ values (corresponding to $\mu_F = 0.2$ and $\mu_F = 0.8$, respectively) obtained from the deterministic settings. It should be noted that we slightly abuse the notation $g_{syn}$ to refer specifically to the maximum synaptic conductance in the double-exponential synaptic model. At each sampled point in the parameter space, we run the simulation $10,000$ times for estimating the mean and variance of the spike count. We should note that we simplify the change of $v_{rest}$ as the membrane reversal potential for passive current due to the large conductance of the leaky channel compared to others. To estimate the energy cost during active and silent signal periods, we integrate the temporal current traces associated with the synapse, the Na$^+$ ion channel, and the leaky Na$^+$ ion channel. The integral of the synaptic current trace is identified as the signal-related energy expenditure, denoted $\epsilon_{sig}$, while the sum of the integrals of the Na$^+$ current from the ion channel and Na$^+$ leak currents is defined as the background energy expenditure, $\epsilon_{bg}$. It is important to note that the majority of the synaptic current is concentrated within the first 100 ms, owing to the 10 ms decay constant used in modeling the synaptic conductance. Additionally, we also convert the integrated currents into ATP by assuming that one ATP molecule is required to pump out three Na$^+$ ions, consistent with the Na$^+$/K$^+$-ATPase pump (Thomas, 1972). Since our simulation only adopts a single leaky channel summarizing all the leaky currents, we use the following formula for estimating the Na$^+$ leak currents:

$$I_{leak}^{\text{Na}^+}(t) = g_{leak}^{\text{Na}^+} \cdot \left(v_{soma}(t) - E_{rev}^{\text{Na}^+}\right), \tag{68}$$

where $v_{soma}(t)$ and $E_{rev}^{\text{Na}^+}$ are the simulated soma temporal potential and Na$^+$ reversal potential, respectively. We further estimate the conductance of the leaky Na$^+$ ion channel, denoted as $g_{leak}^{\text{Na}^+}$, by

$$g_{leak}^{\text{Na}^+} = g_{leak}/(1+r), \tag{69}$$

$$r := \left(E_{rev}^{\text{Na}^+} - E_{rev}^{leak}\right) / \left(E_{rev}^{leak} - E_{rev}^{\text{K}^+}\right), \tag{70}$$

where $E_{rev}^{leak}$ and $E_{rev}^{\text{K}^+}$ denote the leaky channel and K$^+$ reversal potentials.

The above simulation was performed on the distributed cluster computer based on Intel(R) Xeon(R) and does not require any GPU usage. We distributed the simulation of each cell states $(v_{rest}, g_{leak}, g_{leak})$ to a core with individual 1G memory. Each simulation that executes 10,000 run for estimating the mean and variance of spike count takes approximately 40 minutes.

## H    Biophysiological modeling

### H.1    Energy cost

We estimate the total energy cost as:

$$\epsilon_{total} = \kappa \epsilon_{sig} + \epsilon_{bg}, \tag{71}$$

where $\kappa$ represents the level of signal activity. In our simulations, $\kappa$ is varied over the range from 90 to 150. A detailed analysis of how different values of $\kappa$ influence the noise term $\eta_\kappa$ along the optimal adaptation path is provided in Sec. J.

## H.2 TUNING CURVE WIDTH

Following Padamsey et al. (2022), we determine the $\hat{g}_{syn}$ that yields $\mu_F = 0.1$ for each state $(v_{rest}, g_{leak})$. Using these values of $\hat{g}_{syn}$, we define a tuning curve that maps stimulus to synaptic conductance:

$$TC(s; v_{rest}, g_{leak}) = (1 + \phi)\hat{g}_{syn}(v_{rest}, g_{leak}) \, e^{-\frac{1}{2}\left(\frac{s}{170}\right)^2}, \tag{72}$$

where $\phi := 0.02$ is used to compensate for the mean firing rate since $\hat{g}_{syn}$ is determined according to $\mu_F = 0.1$. The stimulus $s$ is defined over the range from $-90$ to $90$ to simulate orientation input. The corresponding tuning curve from stimulus to mean firing rate is obtained by applying $TC$ through the empirically derived mapping from $g_{syn}$ to $\mu_F$ based on simulation data. In contrast to Padamsey et al. (2022), we compute the full width at half maximum (FWHM) of the wrapped curve, as it may deviate from a Gaussian profile. To estimate the width along the optimal path, we fit the FWHM values using a second-degree polynomial function in $v_{rest}$ and $g_{leak}$ and take the estimated values along the optimal states.

## H.3 SPIKE TRAIN STATISTICS

In simulation, we input a deterministic spike and probe the probability of responding spiking output under different configurations, including synaptic conductance, membrane leaky channel conductance, and resting potential. This simulation characterizes the neuron excitability under different conditions. However, profiling this excitability curve does not directly provide the variance of output firing rate required in modeling Fisher information in our framework. To address this limitation, we mathematically derived the variance of output firing rate based on the following assumptions:

- The response output spikings are independent.

- The input spikes follow Poisson distribution with a rate $\lambda \in \mathbb{R}$.

- The variance of the spike count $\sigma_F^2$ can be approximated by a parabola of the mean $\mu_F$ : $\sigma_F^2 \approx \eta_\kappa \mu_F (1 - \mu_F)$.

Based on these assumptions, we define the probability distribution of input spikes $K$ as:

$$p(K = k) = \frac{(\lambda T)^k e^{-\lambda T}}{k!}. \tag{73}$$

In other word, $K \sim Pois(\lambda T)$, where $T$ is a given time window. Since our simulation input is a single input, we can estimate the mean and variance of spike count $F$ by the following empirical distribution:

$$p(F = k; \text{single spike input}) = p_k, \quad k \in \mathbb{N} \cup \{0\}. \tag{74}$$

It should be noted that the number of output spikes $k$ in the simulation is typically less than 3. Then the total output spike counts $O$ is:

$$O = \sum_{i=0}^{K} F_i, \quad \text{(by independence)} \tag{75}$$

where $F_i$ is a realization of $F$ and $F_i$ corresponds to $i$-th spike in the input spiking train. Based on this model, we can calculate the expectation and variance of $O$ by

$$\mathrm{E}[O] = \mathrm{E}[\mathrm{E}[O|K]] = \mathrm{E}[K\mathrm{E}[F]] = \lambda T \, \mathrm{E}[F] = \lambda T \mu_F, \tag{76}$$

$$\mathrm{Var}[O] = \mathrm{E}[\mathrm{Var}[O|K]] + \mathrm{Var}[\mathrm{E}[O|K]] = \lambda T \mathrm{Var}[F] + \lambda T \mathrm{E}[F]^2 = \lambda T \sigma_F^2 + \lambda T \mu_F^2, \tag{77}$$

where $\mathrm{E}[S]$ and $\mathrm{Var}[S]$ can be estimated from the simulation.

Now, we have to approximate the relation between (76) and (77). More specifically, we want to formulate (77) in term of (76).

$$
\begin{aligned}
\frac{\text{Var}[O]}{\text{E}[O]} &= \frac{\lambda T \sigma_F^2 + \lambda T \mu_F^2}{\lambda T \, \text{E}[S]} \\
&= \frac{\lambda T \eta_\kappa \mu_F (1 - \mu_F) + \lambda T \mu_F^2}{\lambda T \, \mu_F} \quad \text{(by } \sigma_F^2 \approx \eta_\kappa \mu_F (1 - \mu_F)) \\
&= \eta_\kappa (1 - \mu_F) + \mu_F \\
&= \eta_\kappa - \mu_F(\eta_\kappa - 1) \\
&\approx \eta_\kappa \quad \text{(if } \mu_F \text{ small)}.
\end{aligned}
\tag{78}
$$

The last approximation holds when neurons operate at the low spiking region, where we make $\mu_F = 0.1$ in our simulation. The eq. (78) also indicates a dispersed Poisson model. In eq. (78), we use a parabolic function to model the variance of spike count. As shown in Fig. 5, the fitted parabolas align closely with the simulation results, demonstrating the accuracy of the approximation.

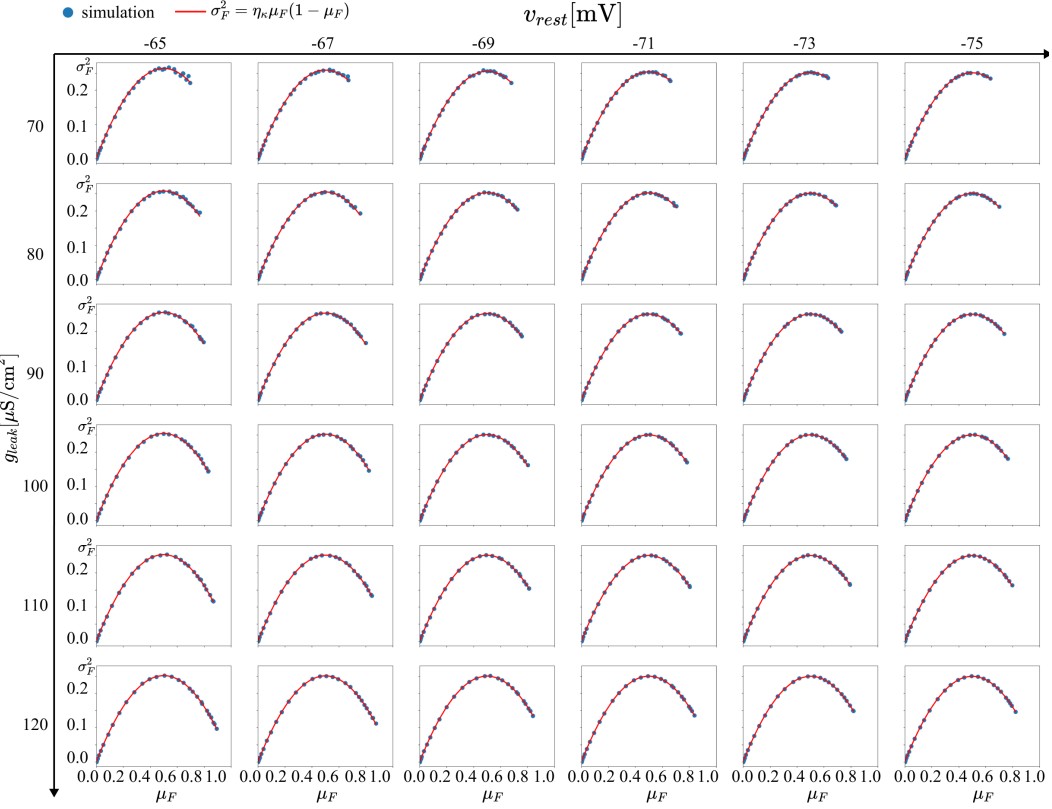

Figure 5: **Fitted parabolas for the mean and variance of spike count.** We fit a parabolic relationship between the mean and variance of spike count across different cellular states $(v_{rest}, g_{leak})$. For visualization purposes, only every second value of $v_{rest}$ and $g_{leak}$ is shown.

## H.4 Fitting of width, energy, and noise

We fit the estimated width and energy in 2 and 1 degree polynomial functions in $v_{rest}$ and $g_{leak}$, respectively (see Fig. 6). Additionally, the estimated noise is fitted with

$$P_4(v_{rest}, g_{leak})e^{w_1 v_{rest} + w_2 g_{leak}}, \tag{79}$$

where $P_4$ denotes the degree-4 polynomial in $v_{rest}$ and $g_{leak}$, and $w_1$ and $w_2$ are the learning weights. Note that we additionally impose a non-negative constraint on the noise fitting parameters, further mitigating potential overfitting. The relative errors of these fittings are less than $2\%$. We select a degree 1 for energy fitting to enhance the stability of the numerical optimization in eq. (15), owing to the linearity of the constraint, which aligns with the straight energy contour. The degrees for the width and noise fitting are determined based on the maximum relative error.

## I Change of energy contours under varying activity level

The change of energy contours under varying $\kappa$ is shown in Fig. 7. Note that underlying noise is independent of $\kappa$ but the angle of constant-energy lines changes, becoming steeper with higher cell activity. This results in different curvatures of the optimal path shape.

## J Noise and width change along the optimal path with varying activity levels

Here, we investigate how the activity level $\kappa$ influences the density (defined as the inverse of tuning curve width) and the behavior of the noise dispersion curve $\eta_\kappa(\epsilon)$ along the optimal adaptations. Specifically, we vary $\kappa$ from 90 to 150 and fit the proposed models to the corresponding density and noise data as described in Sec. 4.1.

The fitted parameters are shown in Fig. 8a-e. To characterize their dependence on $\kappa$, we fit an affine relation to $(a_1, a_2)$ as a function of $\kappa$ (see dotted lines in Fig. 8). Due to the presence of outliers as $\kappa < 100$, we restrict the fitting to data with $\kappa \geq 100$. The resulting fits (dotted lines in Fig. 8a, b) are given by:

$$a_1(\kappa) = -6.193 \cdot 10^{-11}\kappa + 1.462 \cdot 10^{-8}, \tag{80}$$

$$a_2(\kappa) = 1.626 \cdot 10^{-4}\kappa + 0.7452. \tag{81}$$

Additionally, we fit polynomial models of degree 2, 2, and 2 to $b_1$, $b_2$, and $\eta_0$, respectively (see dashed lines in Fig. 8c–e). The fitted expressions are:

$$b_1(\kappa) = 3.093 \cdot 10^1\kappa^2 - 3.922 \cdot 10^3\kappa + 2.013 \cdot 10^5, \tag{82}$$

$$b_2(\kappa) = -5.996 \cdot 10^2\kappa^2 + 1.888 \cdot 10^5\kappa + 3.370 \cdot 10^6, \tag{83}$$

$$\eta_0(\kappa) = -6.102 \cdot 10^{-7}\kappa^2 + 6.004 \cdot 10^{-5}\kappa + 9.934 \cdot 10^{-1}. \tag{84}$$

We further visualize the consequences of these deviations in the resulting density and noise dispersion curves. As shown in Fig. 8f, g, the predicted density and noise dispersion values increasingly deviate from simulation results as $\kappa$ decreases. It should note that the values of $\vec{c}(\kappa)$ can be easily derived from the values of $\vec{a}$ and $\vec{b}$.

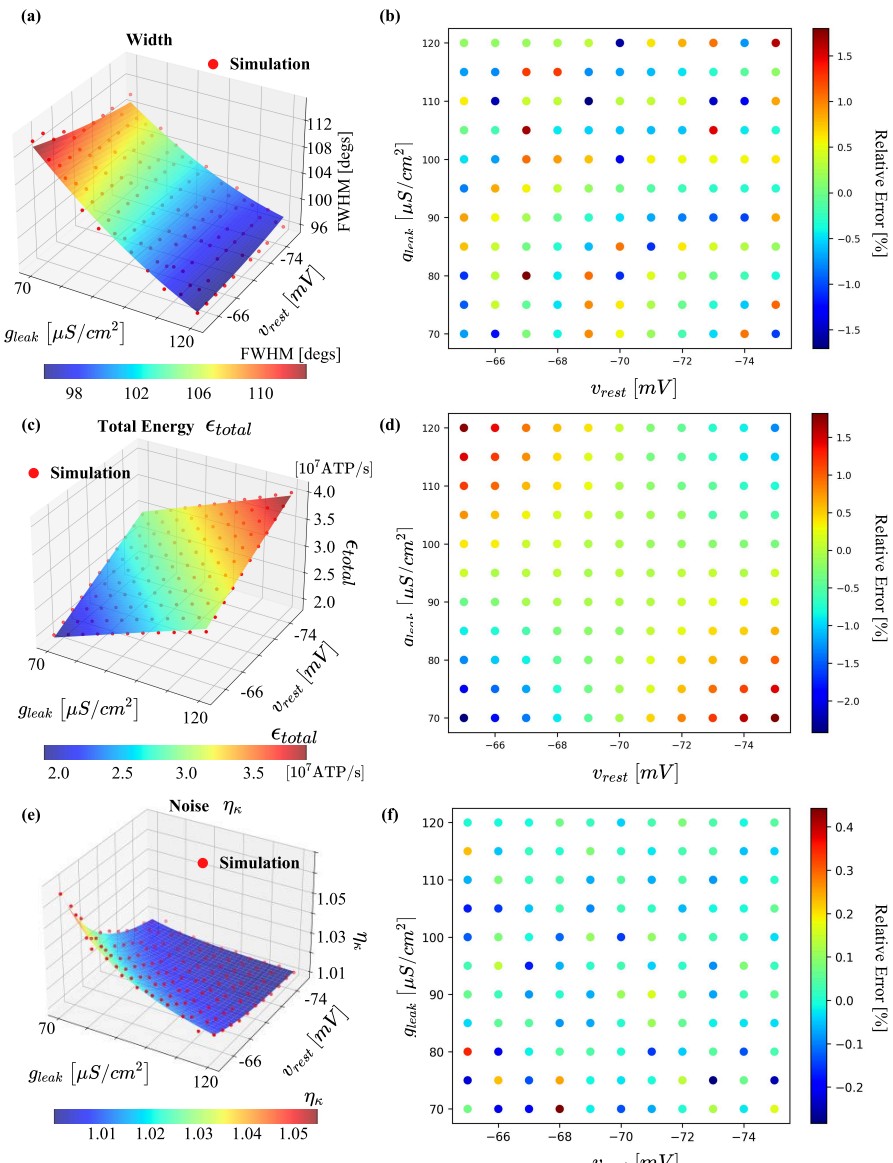

Figure 6: **Surface fitting to width, total energy, and noise.** (a, b) We use 2 degree polynomial function in $v_{rest}$ and $g_{leak}$ to fit the estimated FWHM, with the relative error provided. (c, d) We use 1 degree polynomial function in $v_{rest}$ and $g_{leak}$ to fit the estimated total energy, with the relative error provided. (e, f) We use eq. (79) to fit the estimated total energy, with the relative error provided.

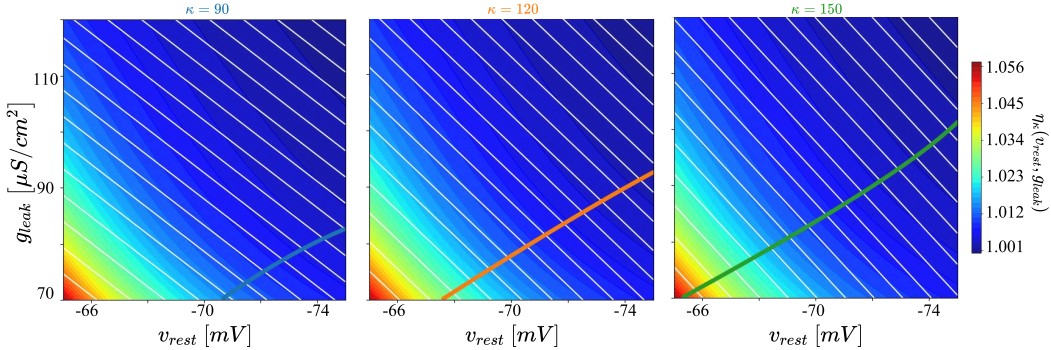

Figure 7: **Impact of cell activity level $\kappa$ on constant-energy contours and optimal paths.** We show varying angles of constant energy contours (white) resulting in different optimal path shapes (blue, orange, green) under $\kappa$ =90, 120, and 150, respectively.

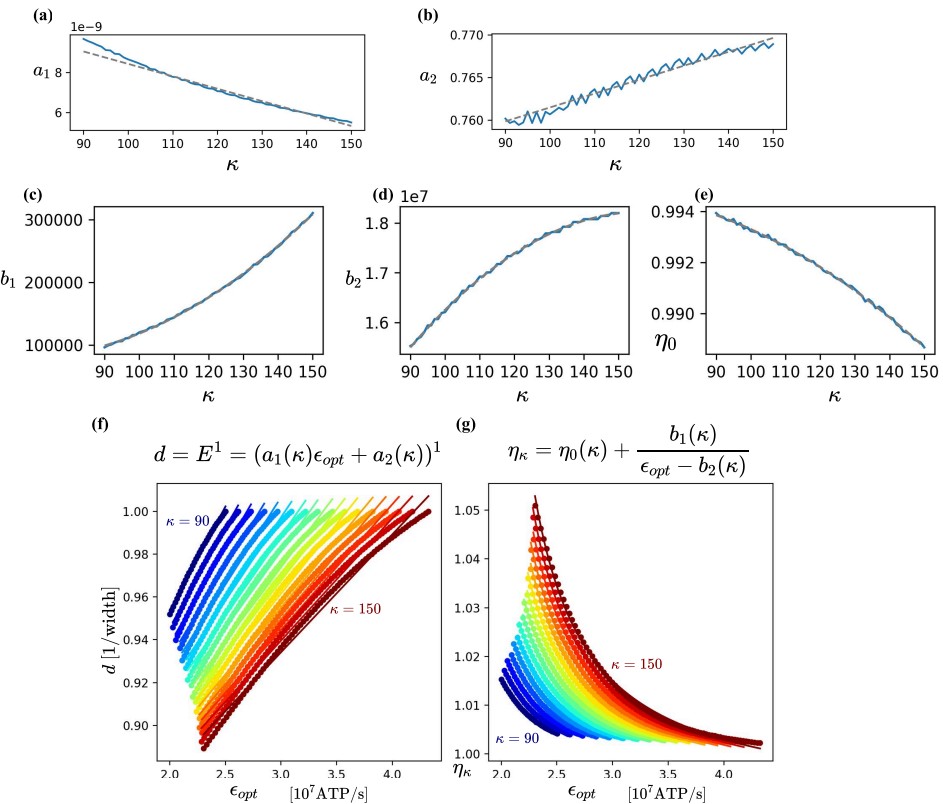

Figure 8: **Relationship between $\kappa$ and the fitting parameters in the energy budget and the noise dispersion model as in eqs. (16) and (18).** (a, b) To represent the energy budget in ATP/s units, we fit the parameters $(a_1, a_2)$ as an affine function of $\kappa$. (c–e) To model the noise in ATP/s units, we fit the parameters $(b_1, b_2, \eta_0)$ with polynomials of $\kappa$. Dotted lines indicate polynomial fits of degree 2, 2, and 2 to $b_1$, $b_2$, and $\eta_0$, respectively. (f, g) The resulting energy budget and noise dispersion curves (solid lines) generated using the fitted expressions are compared with the simulation data. For visualization purposes, we display every fourth value of $\kappa$ in the plotted results.

## K    ANALYTICAL SOLUTION COMPARISON

We compare the our analytical solution to Ganguli & Simoncelli (2010); Wang et al. (2016a) under the different objectives (see. Tabs. 5 to 7).

Table 5: Comparison of infomax analytical solutions

|  | **Ours** | **Ganguli & Simoncelli (2010)** | **Wang et al. (2016a)** |
|---|---|---|---|
| **Density (tuning width)$^{-1}$** $d(s)$ | $E^{\frac{1}{\alpha}}R(s)^{-1}p(s)$ | $Np(s)$ | $Cp(s)$ |
| **Gain** $g(s)$ | $E^{\frac{1}{\alpha}}$ | $R$ | $1$ |
| **Fisher information** $FI(s)$ | $\dfrac{E^{\frac{3}{\alpha}}p(s)^2}{\eta_\kappa(E)\,R(s)^2}$ | $N^2Rp(s)$ | $C^2p(s)^2$ |

Table 6: Comparison of discrimax analytical solutions

|  | **Ours** | **Ganguli & Simoncelli (2010)** | **Wang et al. (2016a)** |
|---|---|---|---|
| **Density (tuning width)$^{-1}$** $d(s)$ | $\propto E^{\frac{1}{\alpha}}R(s)^{\frac{-\alpha-1}{\alpha+3}}p(s)^{\frac{\alpha+1}{\alpha+3}}$ | $\propto Np(s)^{\frac{1}{2}}$ | $\propto Cp(s)^{\frac{1}{3}}$ |
| **Gain** $g(s)$ | $\propto E^{\frac{1}{\alpha}}R(s)^{\frac{2}{\alpha+3}}p(s)^{\frac{-2}{\alpha+3}}$ | $\propto Rp(s)^{\frac{-1}{2}}$ | $1$ |
| **Fisher information** $FI(s)$ | $\propto \dfrac{E^{\frac{3}{\alpha}}p(s)^{\frac{2\alpha}{\alpha+3}}}{\eta_\kappa(E)R(s)^{\frac{2\alpha}{\alpha+3}}}$ | $\propto N^2Rp(s)^{\frac{1}{2}}$ | $C^2p(s)^{\frac{2}{3}}$ |

Table 7: Comparison of general analytical solutions ($L_p$ error, $p = -2\beta$).

|  | **Ours** | **Ganguli & Simoncelli (2010)** | **Wang et al. (2016a)** |
|---|---|---|---|
| **Density (tuning width)$^{-1}$** $d(s)$ | $\propto E^{\frac{1}{\alpha}}R(s)^{\frac{\alpha-\beta}{3\beta-\alpha}}p(s)^{\frac{\beta-\alpha}{3\beta-\alpha}}$ | $\propto Np(s)^{\frac{\beta-1}{3\beta-1}}$ | $\propto Cp(s)^{\frac{1}{1-2\beta}}$ |
| **Gain** $g(s)$ | $\propto E^{\frac{1}{\alpha}}R(s)^{\frac{2\beta}{3\beta-\alpha}}p(s)^{\frac{-2\beta}{3\beta-\alpha}}$ | $\propto Rp(s)^{\frac{2\beta}{1-3\beta}}$ | $1$ |
| **Fisher information** $FI(s)$ | $\propto \dfrac{E^{\frac{3}{\alpha}}p(s)^{\frac{-2\alpha}{3\beta-\alpha}}}{\eta_\kappa(E)R(s)^{\frac{2\alpha}{\alpha-3\beta}}}$ | $\propto N^2Rp(s)^{\frac{2}{1-3\beta}}$ | $C^2p(s)^{\frac{2}{1-2\beta}}$ |

## L    DETAILS IN CONSISTENCY WITH DATA

### L.1    DERIVATION OF THE PARAMETER SETTINGS

In Sec. 5, we aim to find the parameters $\vec{a}$ for simultaneously accomplish 32% widening and 29% reduction in ATP consumption. Combining with the density analytical solution in our model, the

32% widening implies

$$1.32 = \frac{w_{ms}}{w_{ctr}} = \frac{d_{ctr}}{d_{ms}} = \frac{E_{ctr}}{E_{ms}} = \frac{a_1\epsilon_{ctr} + a_2}{a_1\epsilon_{ms} + a_2} \tag{85}$$

$$= \frac{1 + \frac{a_2}{a_1\epsilon_{ctr}}}{\frac{\epsilon_{ms}}{\epsilon_{ctr}} + \frac{a_2}{a_1\epsilon_{ctr}}} = \frac{1 + \frac{a_2}{a_1\epsilon_{ctr}}}{0.71 + \frac{a_2}{a_1\epsilon_{ctr}}} \Rightarrow \frac{a_2}{a_1\epsilon_{ctr}} = 0.19625, \tag{86}$$

where $ms$ denotes metabolic stress.

Given a fixed tuning curve width derived from the fixed population size, we reduce the firing rate according ($R$) in Ganguli & Simoncelli (2010) according to the decrease in the maximum mean firing rate in the mouse neuron. Due to the gaussian assumption, 32% widening, and a uniform prior, we can calculate the decrease in the maximum mean firing rate by

$$\frac{\max_s \frac{1}{1.32\sigma\sqrt{2\pi}}e^{\frac{-s^2}{2(1.32\sigma)^2}}}{\max_s \frac{1}{\sigma\sqrt{2\pi}}e^{\frac{-s^2}{2\sigma^2}}} = \frac{1}{1.32} \approx 76\% = (1 - \underbrace{24\%}_{\text{decrease}}). \tag{87}$$

Similarly, due to 32% widening and a uniform prior, we can calculate the decrease in the coding capacity in Wang et al. (2016a) by

$$\frac{1}{1.32} = \frac{w_{ctr}}{w_{ms}} = \frac{d_{ms}}{d_{ctr}} = \frac{C_{ms}}{C_{ctr}} \Rightarrow C_{ms} \approx (1 - \underbrace{24\%}_{\text{decrease}}) \cdot C_{ctr}. \tag{88}$$

### L.2 MEAN FIRING RATE DEVIATION

Here, we analyze changes in firing rate under different coding objectives and prior distributions (see Fig. 9). Across all coding objectives, we adopt a consistent configuration: $b/(a\epsilon_{ctr}) = 0.19625$, a 29% reduction in energy expenditure, and $\alpha = 1$, consistent with the settings used in Sec. 5. Additionally, we fix $E = 6$ and $R(s) = 1$ for all cases, as these parameters yield a control tuning width of 35 degrees under a uniform prior — following the simulation conditions reported in Padamsey et al. (2022). After generating the population codes according to Tab. 1, we extract the tuning curve corresponding to the neuron with preferred stimulus $s_n = 90°$. Each extracted tuning curve is normalized by the peak value of its respective control condition, a transformation that does not affect the analysis of firing rate deviation.

Fig. 9a shows the uniform prior assumed in Sec. 5, along with the infomax case (Fig. 9b reiterates Fig. 4b). Here we illustrate that, under constant $R(s) = R$, all objective functions lead to the same original tuning curves and adaptations, all with perfect homeostasis.

In addition to the uniform prior, we also consider as a non-uniform prior the distribution of edge orientations in natural scenes, as reported in Girshick et al. (2011). When replacing the uniform prior with the non-uniform one, the deviation in mean firing rate (compared to each control condition, shown in dotted purple) remains within .2% across infomax, discrimax, and $L_1$ error objectives. As shown in Fig. 9g and h, the tuning curves deviate from a Gaussian profile due to the dependence of the gain function $g(s)$ on the prior distribution for $L_p$ norm losses (see Tabs. 6 and 7).

### L.3 MEAN FIRING RATE DEVIATION WITH NON-GAUSSIAN TUNING CURVE

Although we adopt the Gaussian function as our base shape $\hat{h}$, our model is not restricted to it. To further test generality, we evaluate the homeostasis approximation error using a one-dimensional Gabor function, a commonly adopted tuning curve (Manning et al., 2024). As shown in Fig. 10, the mean FR deviation remains within .5%, demonstrating that our homeostasis approximation also holds for non-Gaussian tuning base shapes. Note that the optimality of the derived Gabor tuning curve still requires evaluation of the tiling property, as discussed in Sec. A.

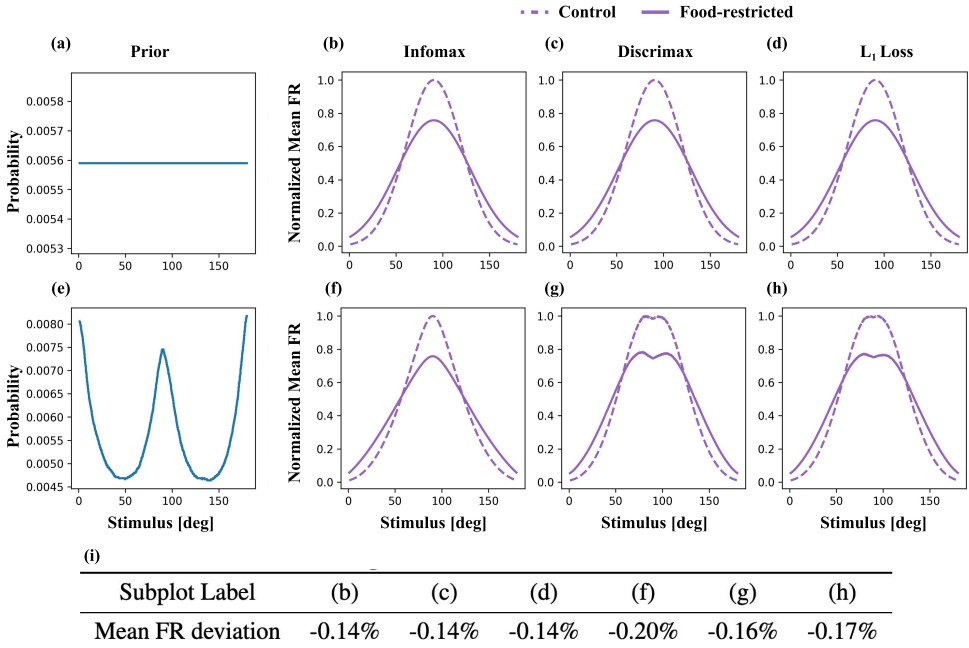

Figure 9: **Gaussian base shape adaptation under varying objectives and a non-uniform prior results in slight violations of homeostasis.** (a) The uniform prior used in Fig. 4. (b) Under a uniform prior and the infomax objective, it reiterates Fig. 4b. (c) Under the discrimax objective, the optimal gain varies as a function of $p(s)$ and $R(s)$. However, by assuming uniform $p(s)$ and $R(s)$, the resulting tuning curve remains unchanged from the infomax case. (d) The $L_1$ error minimization objective yields the same results to (c). (e) A non-uniform prior derived from the distribution of edge orientations in natural scenes, as reported in Girshick et al. (2011). (f) The resulting tuning curve under the infomax objective with the non-uniform prior. (g, h) Under the discrimax and general objectives, the optimal gain varies with $p(s)$ (and $R(s)$, though we assume a constant $R$ in this simulation). Despite these variations, the mean firing rate deviation remains within .2%. (i) The mean FR deviation in the above conditions.

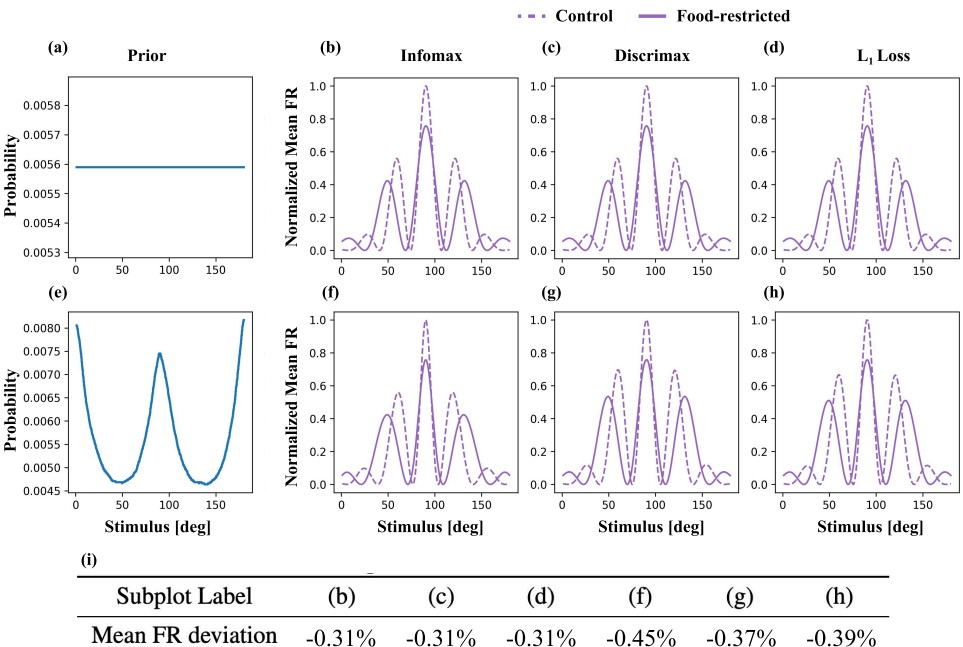

Figure 10: **Gabor base shape adaptation under varying objectives and a non-uniform prior results in slight violations of homeostasis.** (a) The uniform prior used in Fig. 4. (b) Under a uniform prior and the infomax objective. (c) Under the discrimax objective, the optimal gain varies as a function of $p(s)$ and $R(s)$. However, by assuming uniform $p(s)$ and $R(s)$, the resulting tuning curve remains unchanged from the infomax case. (d) The $L_1$ error minimization objective yields the same results to (c). (e) A non-uniform prior derived from the distribution of edge orientations in natural scenes, as reported in Girshick et al. (2011). (f) The resulting tuning curve under the infomax objective with the non-uniform prior. (g, h) Under the discrimax and general objectives, the optimal gain varies with $p(s)$ (and $R(s)$, though we assume a constant $R$ in this simulation). Despite these variations, the mean firing rate deviation remains within .5%. (i) The mean FR deviation in the above conditions.

## M  FITNESS OF THE DERIVED TUNING CURVES

Here, we adopt the likelihood method to evaluate the fitness of our model on the data presented in Padamsey et al. (2022) and compared to Ganguli & Simoncelli (2010); Wang et al. (2016a).

### M.1  LIKELIHOOD

Following the dispersed Poisson distribution, we can derive the log likelihood function of a single neuron as

$$\log \mathcal{L}\left(h_n(s)|r_n^{(1)}(s), r_n^{(2)}(s), \cdots, r_n^{(k)}(s)\right) \tag{89}$$

$$= \log\left[\prod_{j=1}^{k} \frac{1}{\sqrt{2\pi\eta_\kappa(E)h_n(s)}} \exp\left(-\frac{(r_n^{(j)}(s) - h_n(s))^2}{2\eta_\kappa(E)h_n(s)}\right)\right] \tag{90}$$

$$= \sum_{j=1}^{k} \log\left[\frac{1}{\sqrt{2\pi\eta_\kappa(E)h_n(s)}} \exp\left(-\frac{(r_n^{(j)}(s) - h_n(s))^2}{2\eta_\kappa(E)h_n(s)}\right)\right] \tag{91}$$

$$= \sum_{j=1}^{k} \left[-\frac{1}{2}\log\left(2\pi\eta_\kappa(E)h_n(s)\right) - \frac{\left(r_n^{(j)}(s) - h_n(s)\right)^2}{2\eta_\kappa(E)h_n(s)}\right] \tag{92}$$

$$= -\frac{k}{2}\log\left(2\pi\eta_\kappa(E)h_n(s)\right) - \frac{1}{2\eta_\kappa(E)h_n(s)}\sum_{j=1}^{k}\left(r_n^{(j)}(s) - h_n(s)\right)^2 \tag{93}$$

where $k$ is the total number of responses (trials). For optimization, we can drop the constant terms, giving:

$$\log \mathcal{L}_n \propto -\frac{k}{2}\log\left(h_n(s)\right) - \frac{\sum_{j=1}^{k}\left(r_n^{(j)}(s) - h_n(s)\right)^2}{2\eta_\kappa(E)h_n(s)}. \tag{94}$$

To extend the model to the entire population of $N$ neurons, we follow the standard assumption that the noise in each neuron's response is independent. This assumption allows us to formulate the total log-likelihood for the population as the sum of the log-likelihoods for each individual neuron.

$$\log \mathcal{L}_{total} = \sum_{n=1}^{N} \log \mathcal{L}_n. \tag{95}$$

Note that future work could explore the impact of information-limiting noise correlations (Moreno-Bote et al., 2014), but this would require grounding in data that has never been collected to model how these correlations might be impacted by metabolism.

### M.2  CALIBRATION

Estimating the gain function $g$ and density function $d$ by maximizing the log-likelihood in eq. (95) is challenging. The difficulty arises because the second term in the RHS of eq. (94), has a complex, non-linear dependence on $g$ and $d$. Furthermore, the optimization is constrained by the requirements of approximate homeostasis and energy constraint. However, we believe that this method can be used to calibrate the parameters $E$, $\alpha$, and $R(s)$.

$$\log \mathcal{L}_n \propto -\frac{k}{2}\log\left(g(s)\hat{h}(D(s) - D(s_n))\right) - \frac{\sum_{j=1}^{k}\left(r_n^{(j)}(s) - g(s)\hat{h}(D(s) - D(s_n))\right)^2}{2\eta_\kappa(E)g(s)\hat{h}(D(s) - D(s_n))}. \tag{96}$$

Based on the analytical solution (assuming an infomax objective) presented in the manuscript, we can substitute $g(s)$ and $d(s)$ with:

$$g(s) = E^{1/\alpha}, \tag{97}$$

$$d(s) = E^{1/\alpha}R(s)^{-1}p(s). \tag{98}$$

Although eq. (96) is defined in terms of $D$ rather than $d$, we can approximate $D$ numerically. We then can exploit the differentiability of the involved functions to iteratively optimize the parameters using gradient-based approaches. Nevertheless, due to the nonlinear nature of eq. (96), the optimization process may converge to suboptimal local minima.

By the approach above, we can integrate our analytical solution with a data-driven calibration of the parameters $E$, $\alpha$, and $R(s)$. However, the stability and efficacy of this method have not yet been rigorously evaluated. We believe this presents an interesting direction for future work, especially as larger datasets become available.

### M.3    LIKELIHOOD ON BIOLOGICAL DATA

We can apply the method in eq. (96) to evaluate the fit of the tuning curves shown in Fig. 4 to the data in Padamsey et al. (2022). Note that this data has significant drawbacks: rather than taking repeated measurements from the same cell, then repeating the full experiment under a tighter metabolic constraint, Padamsey et al. (2022) instead aligns and averages responses from 29 cells in the control condition and 32 different cells in the food restricted condition, which are taken as representative of variations over repeated trials of a single underlying tuning curve. Because this is the only data available, we will use it as an illustrative example here, but note that future spike train data is needed to refine these estimates. Specifically, we find log likelihoods as -155 (Ganguli & Simoncelli, 2010), -107 (Wang et al., 2016a), -85 (ours), -47 (ours with optimized $\eta$. This shows the expected ordering of fit quality.

## N    EVALUATION OF FLATTENING EFFECT ON NON-INDEPENDENT NOISE

Our proposed framework assumes conditionally independent noise (see Sec. A) for analytical tractability. However, this assumption may be unrealistic due to the presence of noise correlations across neurons. To assess the extent to which such noise model mismatch induces a flattening effect, we incorporate differential correlations (Moreno-Bote et al., 2014), which have been identified as a dominant factor limiting information transmission.

Specifically, we aim to optimize the following constrained optimization problem corresponding to the infomax case:

$$\underset{g(\cdot),\, d(\cdot)}{\operatorname{argmax}} \int p(s) \, \log\left( \frac{\eta_\kappa(E)^{-1} g(s) d(s)^2}{1 + \eta_\kappa(E)^{-1} c\, g(s) d(s)^2} \right) \, \mathrm{d}s, \tag{99}$$

$$\text{s.t.} \qquad p(s)g(s) = R(s)d(s), \tag{100}$$

$$\int p(s)g(s) \, \mathrm{d}s = E, \tag{101}$$

where $c \geq 0$ denotes the strength of differential correlations. In the simulation, we set $\eta_\kappa(E) = 1$, as its value varies only slightly (from 1 to 1.06; see Fig. 3i). For simplicity, we further specify $R(s) = 1$. The stimulus distribution $p(s)$ is chosen to match the empirical distribution of edge orientations in natural scenes reported by Girshick et al. (2011). However, numerically solving this constrained optimization problem is challenging. To mitigate this difficulty, we use the approximate homeostasis constraint to express $d(s)$ in terms of $g(s)$ and relax the energy constraint using an $\ell_2$-norm regularization term with penalty weight $\lambda_{\text{penalty}} = 100$. The relaxed optimization problem is as follows:

$$\underset{g(\cdot)}{\operatorname{argmax}} \int p(s) \, \log\left( \frac{p(s)^2 g(s)^3}{1 + c\, p(s)^2 g^3(s)} \right) \, \mathrm{d}s - \lambda_{\text{penalty}} || \int p(s)g(s) \, \mathrm{d}s - E ||^2. \tag{102}$$

We then perform gradient descent with a step size of $10^{-3}$ for 3000 iterations. After obtaining the optimal $g(s)$, we recover the optimal $d(s)$ by the approximate homeostasis constraint. Note that we vary the correlation strength up to 1000, which is substantially larger than the noise variance of individual neurons (i.e., $\eta_\kappa = 1$).

The optimal gain and density functions exhibit only modest changes across different levels of differential correlations; in contrast, metabolic stress produces a substantially larger effect (see Fig. 11a–b). We then derive the optimal tuning curves associated with the gain and density functions

in Fig. 11a–b, which display the same flattening pattern predicted by our analytical solution in the infomax setting. In Fig. 11d, we quantitatively evaluate the relative broadening for each matched pair of correlation strengths in the control and food-restricted conditions. Although increasing the differential correlation strength from 0 to 1000 results in an additional $4\%$ broadening, this effect is minor compared to the impact of metabolic stress, which produces approximately $38\%$ broadening.

This numerical experiment demonstrates that the flattening effect predicted by our proposed model remains robust even when differential correlations are taken into account.

We conducted similar experiments with $\lambda_{\text{penalty}} = 10$ and $\lambda_{\text{penalty}} = 1000$ (see Figs. 12 and 13) to evaluate whether the broadening effect arises from any numerical artifact introduced by the choice of the penalty weight $\lambda_{\text{penalty}}$. As shown in Figs. 11 to 13, the trend of the broadening effect remains consistent — increasing the differential correlation strength results in additional broadening. We therefore conclude that the broadening effect is not the numerical artifact.

Additionally, we computed the regularizer error,

$$\left\| \int p(s)\,g(s)\,\mathrm{d}s - E \right\|^2,$$

under $\lambda_{\text{penalty}} = 10, 100, 1000$, as shown in Fig. 14. The results show that the default setting $\lambda_{\text{penalty}} = 100$ is a reasonable choice for solving the relaxed problem.

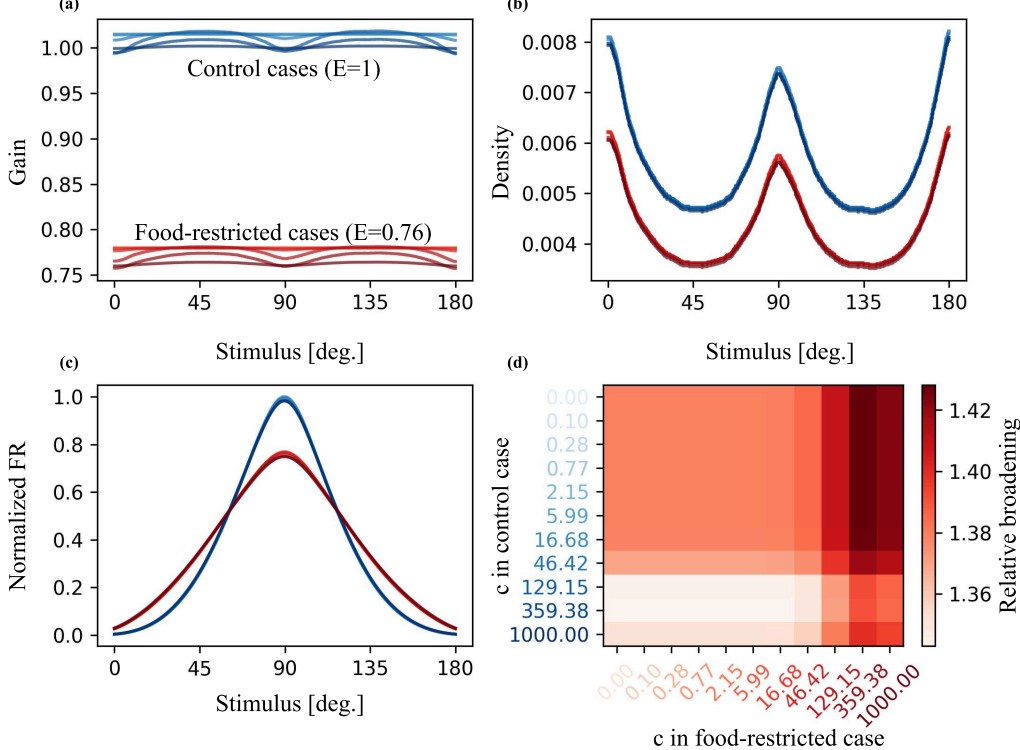

Figure 11: **Comparison of gain, density, tuning curves, and broadening effects under varying levels of differential correlations.** (a–b) Optimal gain and density functions obtained across different strengths of differential correlations. Blue and red curves represent the control and food-restricted conditions, respectively, with color gradients indicating increasing correlation strength. (c) Optimal tuning curves corresponding to the gain and density functions shown in panels (a) and (b). (d) Relative broadening effect for each matched pair of control and food-restricted conditions. The top-left pair corresponds to the case assuming independent noise. Note that this experiment is under $\lambda_{\text{penalty}} = 100$.

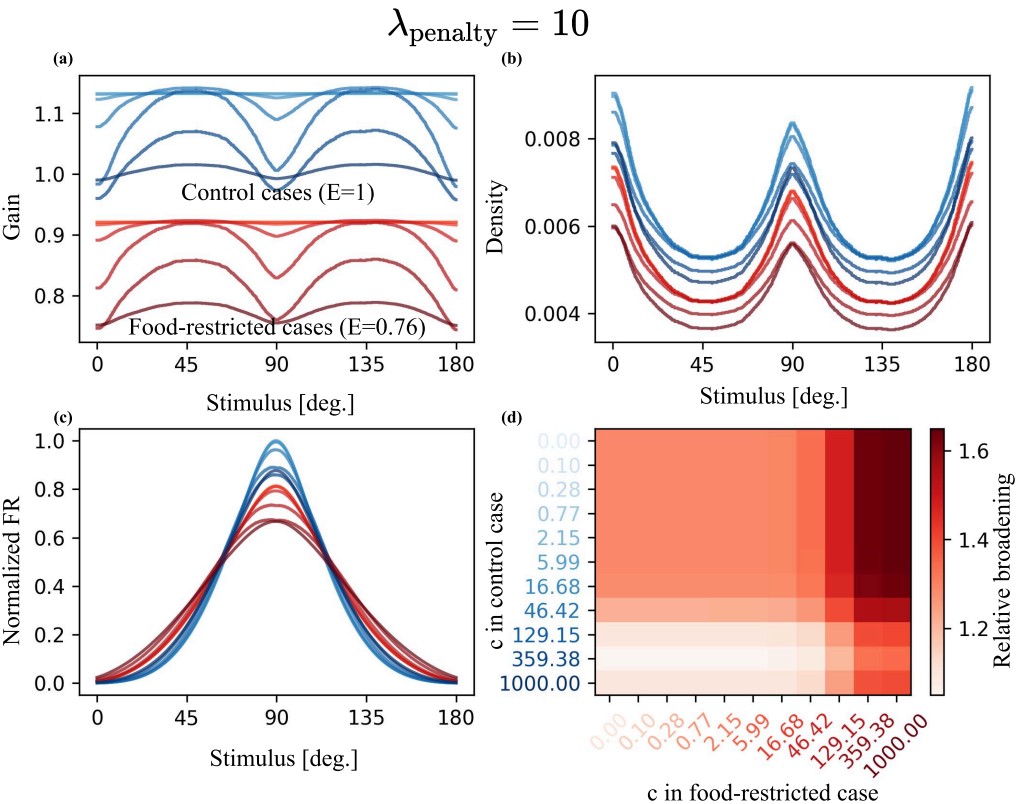

Figure 12: **Comparison of gain, density, tuning curves, and broadening effects under varying levels of differential correlations.** (a–b) Optimal gain and density functions obtained across different strengths of differential correlations. Blue and red curves represent the control and food-restricted conditions, respectively, with color gradients indicating increasing correlation strength. (c) Optimal tuning curves corresponding to the gain and density functions shown in panels (a) and (b). (d) Relative broadening effect for each matched pair of control and food-restricted conditions. The top-left pair corresponds to the case assuming independent noise. Note that this experiment is under $\lambda_{\text{penalty}} = 10$.

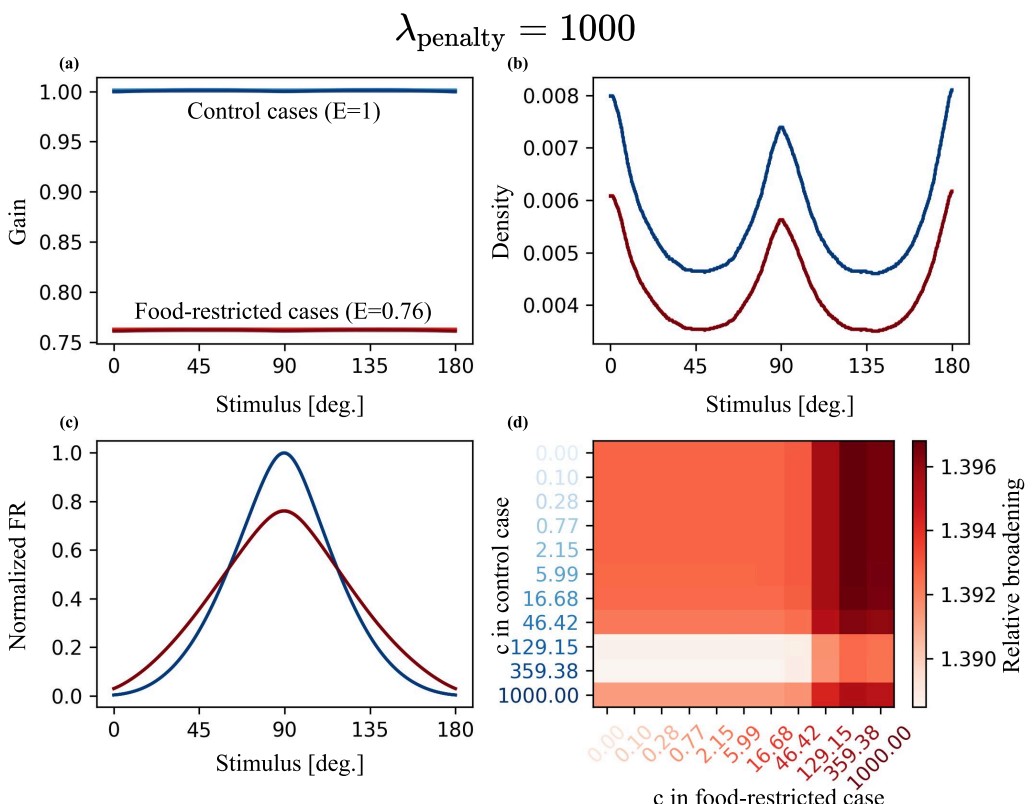

Figure 13: **Comparison of gain, density, tuning curves, and broadening effects under varying levels of differential correlations.** (a–b) Optimal gain and density functions obtained across different strengths of differential correlations. Blue and red curves represent the control and food-restricted conditions, respectively, with color gradients indicating increasing correlation strength. (c) Optimal tuning curves corresponding to the gain and density functions shown in panels (a) and (b). (d) Relative broadening effect for each matched pair of control and food-restricted conditions. The top-left pair corresponds to the case assuming independent noise. Note that this experiment is under $\lambda_{\text{penalty}} = 1000$.

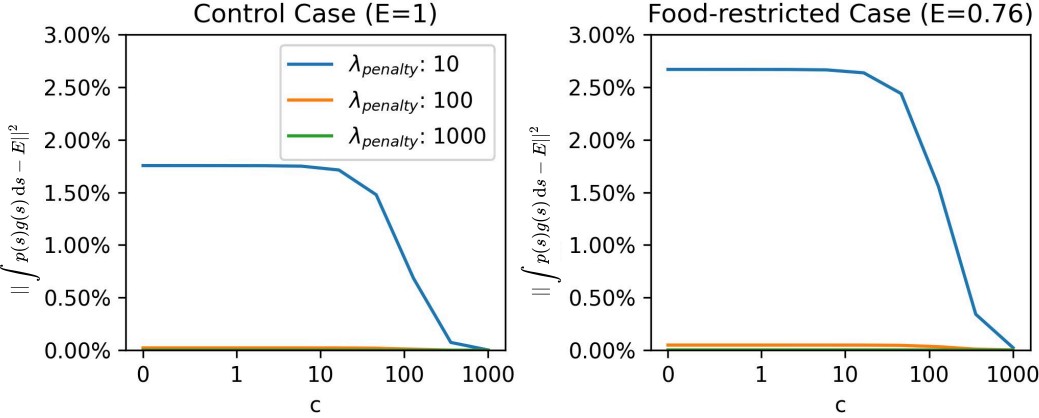

Figure 14: **Evaluation of the effect of penalty weight $\lambda_{\text{penalty}}$.**

