# OpenReview forum: "Homeostatic Adaptation of Optimal Population Codes under Metabolic Stress"
_ICLR.cc/2026/Conference — ICLR 2026 Poster_

### Official Review · Reviewer_rMx9 · 2025-10-29

**Soundness:** 3
**Presentation:** 3
**Contribution:** 3
**Rating:** 6
**Confidence:** 2

**Summary:**

The authors introduce a new method for modelling the interaction of metabolic stress and tuning curves. They show that different from previous models, their model is capable of reproducing the experimental findings reported in (Padamsey et al., 2022) - mainly that under metabolic stress, tuning curves widen and reduce in height. To derive model parameters, they hypothesise that cells adjust neuron parameters (rest potential, leak conductance) to minimise noise levels under varying energy levels.

**Strengths:**

1. The presented model is a simple extension of previous ones, which can be recovered by setting the function eta and parameter alpha accordingly.
2. Analytical solutions are provided for their model for different optimisation objectives.
3. They use simulations inspired by the experimental protocol used in (Padamsey et al., 2022) to fit alpha and eta in a biologically grounded way. This is the main difference between previous models and the proposed one, leading to alpha = 1 (which has also been used in Ganguli & Simoncelli, 2010), and a functional relationship for eta.
4. Their model faithfully reproduces experimental findings. They compare this with two alternative models, which fail to fully capture experimental observations. In particular, only their model shows firing rate homeostasis.

**Weaknesses:**

1. Since the introduced model is calibrated using simulations, it is not clear how well it will generalise to other experimental setups.
2. In general, the part about extracting eta_k and alpha from simulations is quite dense and hard to follow, which could be improved.
3. Some of the used variables have to be explained better, as it is not evident what they are or how they are determined (see questions).

**Questions:**

1. As far as I can see, the main novelties are: 1) The activity and energy-dependent dispersion, 2) The generalised energy constraint, although in the derived model, alpha = 1 is used which aligns with previous models. 3) The fit to simulated results (with energy estimates tying neural activity to ATP needs) to  obtain alpha and eta. Is that correct?
2. Eq. (1) is for Poisson neurons? A citation would be helpful.
3. What exactly is kappa? How is it related to R(s), R_n, R, etc.? Also, how is R(s) determined?
4. Eq. (7): E is stated to be energy. However, since this expression is valid for any alpha >= 1, this is weird unit-wise (and in fact, for alpha = 1, the authors identify E as the mean rate R).
5. The final model is more complex than the two comparison models (Ganguli & Simoncelli; Wang et al.), and parameters are actually fit - not on experimental data, but using simulations. So it seems like increased expressivity is paid for by an increase in complexity. Could you comment on how well you expect this to generalise to other experimental setups? In particular, does your model predict any other effects?
6. If I understand correctly, eta_k is obtained for different combinations of v_rest and g_leak. For activity kappa and a given energy level, you then determine the values of (v_rest, g_leak) that minimise eta_k (which is later used to fit a function for eta_k)? This part was quite dense to read, so a few more explanations might be helpful here.

---

> ### Author Response · Authors · 2025-11-21
> **W1: Generalization to other experimental setups**
>
> Thank you for raising this concern. We would like to reiterate that our proposed framework encompasses the competing models (Ganguli \& Simoncelli, 2010; Wang et al., 2016) as special cases. As a result, it preserves their explanatory power and, in particular, matches the ability of Ganguli \& Simoncelli (2016) to account for the empirical data across the five measured attributes (three visual and two auditory). The generalization of our proposed model to other experimental conditions—particularly those involving metabolic stress—remains difficult to assess due to the limited availability of suitable datasets. To the best of our knowledge, the data reported by Padamsey et al. represent the only publicly accessible dataset in this regime, and our framework provides the closest fit to these observations (see Fig. 3 and Sec. M).

---

> ### Author Response · Authors · 2025-11-21
> **W2 & Q6: Explanation of eta_k and alpha**
>
> Thank you for providing this feedback. We updated Lines 432-439 to improve the clarity. The conclusion that $\alpha = 1$ follows from the definition of the metabolic budget $E$, expressed in units of ATP/s as
> \begin{align}
> E(\kappa) &= a_1(\kappa)\epsilon_{\mathrm{opt}} + a_2(\kappa).
> \end{align}
> From Table. 1, the optimal density scales as
> \begin{align}
> E^{1/\alpha} &= \left(a_1(\kappa)\epsilon_{\mathrm{opt}} + a_2(\kappa)\right)^{1/\alpha}.
> \end{align}
> Because Fig. 2h demonstrates an approximately affine relationship between density and $\epsilon_{\mathrm{opt}}$, this scaling is consistent when $\alpha = 1$. The corresponding goodness-of-fit results under this choice of $\alpha$ are provided in Fig. 8f.
>
> The calibration of $\eta_\kappa$ is obtained from the simulated noise strength evaluated along the optimal path (Fig. 2i). Specifically, we numerically solve eq. (15) using the simulated data; after determining the optimal path for a given ATP/s level, we extract the noise strength along this path and fit it using eq. (18). Details of this fit are shown in Fig. 8g.

---

> ### Author Response · Authors · 2025-11-21
> **W3: Variable definitions**
>
> Thank you for proving this feedback. We included Table. 4 for the variables used in the framework. Additionally, we included Table. 3 to summarize the parameters used in simulation. We moved the simulation pipeline (now Fig. 2) into the manuscript for improving the simulation clarity.

---

> ### Author Response · Authors · 2025-11-21
> **Q1: The main novelties**
>
> Yes, your understanding is correct. However, we would like to reiterate that the approximate homeostasis constraint constitutes one of the key novelties of our framework, as it has not been employed in prior work. Although the choice $\alpha = 1$ is consistent with the energy constraint used in Ganguli \& Simoncelli (2010), to the best of our knowledge we are the first to calibrate this parameter through simulation and to tie it directly to metabolic energy rather than to firing rate, which serves only as a proxy for energy.

---

> ### Author Response · Authors · 2025-11-21
> **Q2: Equation 1**
>
> This result holds for both Poisson and Gaussian neurons. We provide the proof for the dispersed Poisson case in Sec. B, and the derivations for the Poisson and Gaussian neurons follow analogous steps. The Gaussian with a constant variance case can be found in Wang et al., (2016).

---

> ### Author Response · Authors · 2025-11-21
> **Q3: Kappa, R(s), R_n, and R**
>
> The parameter $\kappa$ represents the level of signal activity. In our simulation, we estimate the energy cost of sending and receiving signals (denoted as $\epsilon_{sig}$) and during silent periods (denoted as $\epsilon_{bg}$); see Sec. G for details. To address the limitation of our single-compartment simulation with a single spike input, we introduce $\kappa$ to approximate the total energy expenditure as
> \begin{align}
>     \epsilon_{total} = \kappa\, \epsilon_{sig} + \epsilon_{bg}.
> \end{align}
> Intuitively, a neuron with higher activity corresponds to a larger value of $\kappa$, leading to greater energy consumption associated with spiking activity. For any given real neuron, this scaling would need to be determined empirically from a measurement of presynaptic currents and other neuronal parameters. In addition to $\kappa=120$, which is selected to match observations by Padamsey et al. (specifically: start and end resting potential and percent change in leak conductance). We additionally consider an extended range of values ($\kappa \in \{90,120,150\}$ vs component ranges of $\epsilon_{sig} \in [62260, 316695] $ and $ \epsilon_{bg} \in [3939771, 10420693]$ in a unit of ATP/s) to provide a more general model for anticipated future neural data.
>
> Ganguli \& Simoncelli (2010) use the parameter $R$ to denote the total expected firing rate of a neuron population. The parameter $R_n$ represents the expected firing rate of the $n^{\mathrm{th}}$ neuron in the population. Meanwhile, $R(s)$ can be interpreted as the expected firing rate of a neuron whose preferred stimulus is $s$, and can be regarded as a "continuous" version of $R_n$.
>
> In our simulations, we set $R(s) = 1$ for simplicity. Determining the actual values of $R(s)$ would, however, require a series of neuronal experiments to measure or estimate $R_n$, followed by a procedure to continuize $R_n$ into $R(s)$.

---

> ### Author Response · Authors · 2025-11-21
> **Q4: Unit of E**
>
> Eq. (7) incorporates the parameter $\alpha$ into the proposed energy constraint due to the following reasons. We found that when $\alpha = 1$ and the firing rate $R$ is used as a proxy for energy $E$, the formulation recovers the energy constraint described in Ganguli \& Simoncelli (2010). In contrast, setting $\alpha = 1.5$ allows the framework to recover the coding capacity constraint used in Wang et al. (2016). We acknowledge that in these cases, the calibrated constants described in Sec. J will have different physical units.
>
> We would like to emphasize that we do not equate $E$ directly with the mean firing rate $R$. Rather, we show that when this incorrect simplification is made, our general case reduces to the naive energy constraint in Ganguli \& Simoncelli (2010) rather than our properly characterized ATP-based constraint.

---

> ### Author Response · Authors · 2025-11-21
> **Q5: Model expressiveness**
>
> In our general case, the added complexity of our model is the addition of $\alpha$, $\eta_\kappa(E)$, and $R(s)$. We remove $\alpha$ in the special case by calibrating it to 1 with biophysical simulation. It is true that $\eta_\kappa(E)$ and $R(s)$ require additional data-driven grounding for specific populations, but we believe that introducing these extra parameters are valuable since it narrows down the information-relevant quantities. Most importantly, these quantities can be measured and identify the useful additional complexity.
>
> In addition to expressing the observed widening of real tuning curves, our model makes the following predictions (see Table 1):
>
> - The tuning-curve width scales on the order of $E^{-1}$.
> - The tuning-curve width is robust with respect to the neuronal noise variance $\eta_\kappa$.
> - The discrimination bound scales on the order of $E^{-3/2}$.
> - The discrimination bound scales on the order of $\sqrt{\eta_\kappa}$.
>
> Note that these predictions hold for infomax, discrimax, and general cases. It also gives a new test for the objective of a brain region (Manning et al., PLoS computational biology 2024).
>
> Our prediction of the discrimination bound is derived from the Cramér–Rao bound, which can be attained by an unbiased efficient decoder. More specifically, under the assumption of a downstream unbiased and efficient neuronal decoder, the discrimination bound can be interpreted as the minimal stimulus difference that can be reliably discriminated at a given stimulus value.
>
> It is also worth noting that validating these predictions is challenging, as the effects of metabolic stress on single neurons and neuronal networks remain underexplored. However, we believe that the central motivation of this work — understanding how metabolic stress shapes neural coding — can be further advanced by future experimental investigations that directly probe metabolic constraints alongside neural response properties.

---

### Official Review · Reviewer_LgZi · 2025-10-29

**Soundness:** 3
**Presentation:** 2
**Contribution:** 2
**Rating:** 4
**Confidence:** 3

**Summary:**

The paper tackles the adaptation of tuning curves in optimal networks in situations of metabolic stress. Authors model the observation that under metabolic stress, tuning curves of neurons in the mouse visual cortex show flattening while the average firing rates are maintained. A theoretical framework describing tuning curves is developed that captures such flattening of tuning curves in situation of energy limit that affects noise levels in single neurons. Simulation of biophysical neurons are carried out to show that the mechanism suggested by the theory can be also seen in the biophysical model of single neurons.

**Strengths:**

The theoretical framework as well as numerical simulations seem technically sound and carefully conducted (even though I had not checked all the details). The question attacked by the paper is  seldom explored and is of biological relevance. Some of the relevant weaknesses are appropriately discussed.

**Weaknesses:**

1) Main weakness is the assumption of conditionally independent firing rates. Cortex, including the visual cortex of the mouse, consists of highly recurrently connected networks of Excitatory and inhibitory neurons that have strong influence on each-other's activity (see for example Chettih and Harvey, Nature 2019). I am not convinced that the assumption of independent neurons can bring crucial insights about the neural code in the cortex, including the study of tuning curves. Can authors comment why do they think that a model with independent neurons is relevant?

2) There is a lack of citations on several occasions.
Examples:
[line 50 ] Authors mention "Simple models of energy constrains" but not cite any so it remains unclear which models they are referring to.
[line 54] ..."previous population coding models capture either the shortening or the widening of energy-limited tuning curves, but not both". Which previous models show this and why are they not cited at this point?

3) Authors situate the current work among the existing literature in a way that seems rather problematic. In line [122] authors comment on the importance of the activity of the sodium/potassium pump for the metabolic expenditure of neurons. The activity of the sodium/potassium pump seems to be a direct consequences of spiking activity, and the maintenance of the reversal potential also depends on neuron's firing rate. It can be argued that models that formulate the metabolic cost that is dependent on the firing rate or the number of fired spikes nevertheless capture, even though indirectly, sources of metabolic expenditure that are biologically highly relevant. Such efficient models simply do not model the situation of important metabolic stress, but instead use the metabolic cost as a means to regulate the activity levels in the network and constrain the solution appropriately (in a general situation without energy deprivation). While a state of energy deprivation seems an important constraint from the  evolutionary perspective, I am not convinced that neural coding would only adapt to this particular constraint.

Recent research (Gutierrez and Deneve, eLife 2019, Koren et al., eLife 2025) has discussed the role of metabolic efficiency in optimal spiking networks. Both studies use a metabolic constraint that is formulated as a cost on the number of spikes fired. The first study (Gutierrez et al.) found a formulation of efficiency that gives rise to a transient adaptation in single neurons on a time scale of seconds. The second study (Koren et al.) found that a number of biophysical parameters in primary cortex can be captured when the metabolic efficiency is taken into account, and that metabolic efficiency was shown to have an important effect on optimal coding solutions. Can authors comment on their results in light of these two studies and also touch on the difference in the use of the metabolic efficiency in their model compared to the one in papers mentioned above?

4) The paper is difficult to read and could use a rewrite with focus on clarity. I believe bringing in some more simplicity when possible would make the paper much more appealing. In particular, the first paragraph of discussion gives a good intuitive motivation for the model and I suggest to move this part earlier in the paper, possibly to the introduction.

**Questions:**

1) What kind of neuron model is used in simulations? I believe it is a Hodgkin-Huxley model, but this is not specified.
2) In line [300], authors mention mean spike count of 0.8 and 0.2. It is unclear to me what are the units used here?

---

> ### Author Response · Authors · 2025-11-21
> **W1: Main weakness of conditionally independent firing rates**
>
> Thank you for bringing attention to the critical gap between theoretical modeling and empirical evidence. Here, we demonstrate that our theoretical results, as summarized in Table. 1, remain relatively robust when incorporating correlated noise among neurons. To facilitate this analysis, we adopt the framework of differential correlations (Moreno-Bote et. al., Nature Neuroscience 2014), which has been shown to be the dominant factor in limiting information. Notably, these correlations implicitly reflect the positive noise correlations observed between neurons with similar tuning preferences (Chettih and Harvey, Nature 2019). Additionally, the following results are updated in Sec. N with Fig. 11.
>
> More specifically, we model the FI with the differential correlations as a function of stimulus $s$ and energy $E$ as:
>
> $FI_{\text{diff}}(s; E) = FI_{\text{ind}}(s; E) - \frac{FI_{\text{ind}}^2(s; E)}{1 + c FI_{\text{ind}}(s; E)} = \frac{FI_{\text{ind}}(s; E)}{1 + c FI_{\text{ind}}(s; E)},$
>
> where $FI_{\text{diff}}(s; E)$ and $FI_{\text{ind}}(s; E)$ denote the FI assuming differential correlations and conditionally independent noise (see Sec. A in supplement), respectively. The non-negative constant $c$ represents the strength of the noise correlations.
>
> However, numerically solving the constrained optimization problem, constituted by the above $FI_{\text{diff}}$ and energy and approximate homeostasis constraints in our framework, is challenging. To mitigate this difficulty, we use the approximate homeostasis constraint to express $d(s)$ in terms of $g(s)$ and relax the energy constraint using an $\ell_2$-norm regularization term with penalty weight $\lambda_{\text{penalty}} = 100$. The relaxed optimization problem is as follows:
>
> $argmax_{g(\cdot)} \int p(s) \log \left(\frac{p(s)^2 g(s)^3 }{1 +  c\, p(s)^2 g^3(s) }\right) \mathrm{d}s - \lambda_{\text{penalty}} ||\int p(s) g(s) \mathrm{d}s - E||^2.$
>
> It should note that this relaxation allows a mild violation of the energy constraint.
>
> Fig. 11a-c illustrate that the noise strength (indicated by the color gradient) exerts a relatively minor effect compared to the impact of energy constraints (with blue corresponding to the control group and red to the hungry group). Furthermore, Fig.~11d shows the broadening effect across different pairs of noise strengths in the control and hungry conditions. These results suggest that a larger noise gap between the hungry and control groups ($c_{\text{hungry}} - c_{\text{control}}$) tends to increase the broadening effect. Although the noise gap increases the broadening effect, it does not qualitatively change the trend of broadening effect from the control case to the hungry one. Specifically, in the fully independent noise case, our model predicts a broadening of 38% and in the most extreme case (control noise is independent (c=0), hungry noise has $c=359.38$), we predict an only slightly modified broadening of 43%.
>
> It should be noted that an analytical solution for the differential correlation case is not available; therefore, we perform the analysis numerically.

---

> > ### Comment · Reviewer_LgZi · 2025-11-27
> > **additional minor questions**
> >
> > Thank you for addressing this important caveat. I have two minor questions.
> >
> > 1) Conditionally independent noise - is this meant relative to the stimulus?
> >
> > 2) How is penalty weight chosen? Does the result vary a lot if a different penalty weight is chosen?

---

> ### Author Response · Authors · 2025-11-21
> **W2: Lack of citations**
>
> Thank you for pointing out this issue. We modified the manuscript as follows:
>
> - **(Line 56)** Simple models of energy constraints (e.g., on mean firing rate in Ganguli \& Simoncelli (2010); Yerxa et al. (2020); Rast \& Drugowitsch (2020) or maximum firing rate in Wang et al. (2016a;b); Laughlin (1981)) are analytically convenient, but they fail to capture the complex trade-offs determining energetic costs in real cells, such as subthreshold activity and biophysical adaptations (Padamsey \& Rochefort, 2023).
>
> - **(Line 64)** Specifically, our model predicts tuning curve flattening closely in line with Padamsey et al. (2022), while previous population coding models capture either the shortening (Ganguli \& Simoncelli, 2010) or widening (Wang et al., 2016a) behavior of energy-limited tuning curves, but not both.
>
> The detail comparison of our model and others is summarized in Tab. 2. We will also reference this table to the necessary locations for improving the clarity.

---

> > ### Comment · Reviewer_LgZi · 2025-11-27
> >
> > Thank you for providing these citations - I believe that they clarify the significance of your contribution.

---

> ### Author Response · Authors · 2025-11-21
> **W3: Critique of energy constraint**
>
> Thank you for highlighting this thought-provoking perspective, and we have included the suggested works and others (Tavoni, bioRxiv 2025; Koren \& Panzeri, NeuroIPS 2022) in our paper (see lines 141-142).
>
> We would like to point out that using firing rate as a proxy for energy consumption cannot capture the measured effects of metabolic stress. While total energy expenditure can be approximated as the product of energy per spike and firing rate—so firing rate serves as a reasonable proxy when the per-spike energy cost is constant—this assumption is not always valid. Padamsey et al. (2022) showed that under metabolic stress, **spike rates remain the same while energy use is reduced**. This "low power mode" cannot be described accurately with a spike-count energy cost. Our model accurately captures the biophysical trade-offs in per-spike cost and information lost to noise, reflecting the fact that not all spikes are equally expensive. Specifically, the amount of ion exchange required per spike can vary significantly. We have added text following lines 140-147 to clarify this point.
>
> We recognize that neural coding involves multiple interacting factors, and energy constraints are a major component. Although our model does not aim to explain the full complexity of neural coding, it establishes a direct link between model parameters and energy consumption (for general Lp loss functions), providing a more accurate representation of metabolic influences. Most importantly, this energy-based formulation does not conflict with prior models; instead, it generalizes them by making their assumptions explicit in terms of energy (see Fig. 1).

---

> > ### Comment · Reviewer_LgZi · 2025-11-27
> >
> > Thank you, this revision is also relevant and very helpful in clarifying how your approach differs from other approaches addressing the topic of metabolic efficiency in neural networks.

---

> ### Author Response · Authors · 2025-11-21
> **W3.5: Comments on recent research works**
>
> Thank you for sharing these inspiring works. The two studies highlighted share an important perspective: the role of metabolic constraints in neural coding. They explain adaptation and network-level neuronal characteristics by approximating energy expenditure through firing rate. This perspective provides a valuable direction for future work, as discussed in our discussion section. Specifically, the observations reported in Padamsey et al. (2022) and the simulations in our study are based on single neurons and extending these findings to a network-level framework represents an important next step.
>
> We note that both aforementioned studies do not account for changes in intrinsic neuronal properties. Thus, adaptations to long-term caloric deprivation, encompassing intrinsic cell-level and/or network-level properties, remain an open question requiring further investigation.
>
> Additionally, we compare the models of Gutierrez and Deneve, eLife 2019 and Koren et al., eLife 2025 with those in Ganguli \& Simoncelli (2010), Wang et al. (2016), and our own work. The biologically plausible models in the former set treat energy expenditure as a regularizer, whereas the latter set formalizes energy-associated constraints (i.e., firing rate or metabolic stress) as a hard constraint. In this sense, the former can be considered a “soft” constraint version of the latter. While the soft-constraint formulation is appealing, it introduces an additional parameter—the penalty constant for information and energy—which is difficult to measure biologically. Consequently, we argue that incorporating energy as an explicit constraint is more practical for both biological and theoretical analyses.
>
> We would like to emphasize again that our model generalizes previous models that use firing rate as an energy proxy, building upon their results without challenging their validity.

---

> ### Author Response · Authors · 2025-11-21
> **W4: Paper clarity**
>
> Thank you for proving this valuable feedback. We modified the first paragraph of the Discussion and updated it to the Introduction section (Lines 44-51) for emphasizing the motivation and clarity of this work.

---

> ### Author Response · Authors · 2025-11-21
> **Q1&2**
>
> Yes, we used a stochastic extension of the Hodgkin-Huxley model, with a mean spike count of 0.8 and 0.2 total during the 2 second simulation duration. We prefer to report spike totals rather than a rate in Hz, as most activity is observed in the first 100 ms, so that rates over 2 s can be misleading.
>
> We updated the manuscript as follows:
>
> - **(Line 327)** We simulate a single-compartment neuron based on a stochastic Hodgkin-Huxley model in response to an input spike, under a variety of settings.
> - **(Lines 335-337)** This range allows us to exclude cell parameter triplets that lead to mean spike count over .8 or under .2 total during the 2 second simulation duration in the deterministic setting, so that stochastic simulation captures meaningful variance.
>
>
> Full NEURON code is available in the supplement.

---

> ### Comment · Reviewer_LgZi · 2025-11-27
>
> Thank you for clarifying this.

---

> ### Author Response · Authors · 2025-11-27
> **Rebuttal Q1&2: conditionally independent noise and the selection of penalty weights**
>
> Thank you for the feedback. These questions help us improve the paper.
>
> 1. The conditional independent noise assumption indicates that the noise for each neuron is independent when conditioned on the stimulus $s$ and the energy level $E$. More specifically, each neuron fires according to $\mathcal{N}(h_n(s), \eta_\kappa(E) h_n(s))$. It should be noted that this does not mean we exclude the variance arising from the stochastic response of spikes to a single input spike. That source of variance is captured through the stochastic Hodgkin-Huxley model and is characterized via the parabolic fitting (see Fig. 3e and Fig. 5).
>
> 2. We included additional experiments to evaluate whether the choice of the penalty weight is reasonable and to confirm that the observed broadening effect does not arise from this choice. As shown in the newly added Figs. 12-14, we conclude that the broadening effect is not a numerical artifact and that the default penalty weight $\lambda_{\text{penalty}} = 100$ is a reasonable setting.

---

> ### Comment · Reviewer_LgZi · 2025-11-28
> **update of the Presentation and Contribution Score and the general Rating**
>
> The revision has successfully generalised the result to a biologically even more relevant setting by including the analysis of differential correlations - this has suitably addressed my main concern. My secondary concerns about text clarity have also been well addressed by the authors. In light of these improvements, I rise my score for Presentation to 3, my score for Contribution to 3, and the general Rating to 8.

---

### Official Review · Reviewer_iz34 · 2025-10-30

**Soundness:** 3
**Presentation:** 3
**Contribution:** 3
**Rating:** 6
**Confidence:** 4

**Summary:**

The idea that brain representations are fundamentally shaped not only by input statistics but also on metabolic constraints has proven very important for our understanding of sensory coding. Nonetheless traditional efficient coding models don;t capture the exact nature of the constraints which leads to incorrect predictions in the low energy regime compared to recent experimental measurements (Padamsey et al., 2022). The paper introduces a new way of expressing these energetic constraints that is mathematically tractable and biophysically well justified, part of a general framework that has past model as special cases, and accounts for the experimental observations that demonstrate increased noisiness as the price of more stringent energy constraints.

**Strengths:**

Clean mathematical formalism for coding efficiency with new constraints.

Biophysical simulations that link mechanistic considerations with coding level abstraction.

Explains for the first time recent experimental observations on the effect of limited energy availability on neural coding.

**Weaknesses:**

While i do enjoy the mathematically clean formulation, the change in the constraint is in and of itself an incremental contribution at the technical level.

The link to data and discussion sections are very brief and need expansion.

Numerical results are very minimal.

**Questions:**

Neural results and discussion: i would have liked to see some concrete predictions that the model makes and perhaps some comments of how the energy-dependent encoding process affects processing downstream

some additional validation of results in simulations, e.g. documenting how other coding considerations affect the degree of flattening would make the coding numerics more substantial.

---

> ### Author Response · Authors · 2025-11-21
> **W1: Incremental contribution at the technical level**
>
> Thank you for raising this concern. We would like to clarify that the approximate homeostasis (AH) constraint introduces several points of novelty:
> 1. To our knowledge, the AH constraint has not been previously formulated or applied to population-level neural coding problems, which represents a technical contribution.
> 2. The AH constraint provides a principled way to generalize existing models (see Fig. 1).
> 3. The AH constraint enables a transparent interpretation of the parameter $\alpha$; for example, $\alpha = 1$ and $\alpha = 1.5$ correspond to energy and coding-capacity constraints, respectively.
>
> Together, these contributions allow our model, unlike any in the existing literature, to accurately capture the behavior of food-restricted neurons observed by Padamsey et al.

---

> ### Author Response · Authors · 2025-11-21
> **W2: Brief link between data and discussion sections**
>
> Thank you for providing this valuable feedback. The result sections span across Secs. 4 and 5. In Sec. 4, we use the stochastic Hodgkin-Huxley based model to calibrate the parameters of $\alpha$ and $\eta_\kappa$; in Sec. 5, we show that the solution to our own model can predict the flattening effect in the tuning curve (Padamsey et al., 2022). We highlighted these results in the discussion section as follows in Lines 510-518.

---

> ### Author Response · Authors · 2025-11-21
> **W3: Minimal numerical results**
>
> Thank you for pointing out this weakness. Due to the page limit, we organized some of the numerical results in the supplement. The summarized organization of the numerical results is as follows:
>
> - (Sec. H4) The fitting error of width, energy, and noise.
> - (Sec. L2) Evaluation of AH constraint efficacy in the Gaussian setting.
> - (Sec. L3) Evaluation of AH constraint efficacy in the Gabor setting.
> - (Sec. M) Fitness of the derived tuning curves on biological data.
> - (Sec. N) Evaluation the robustness of the analytical solution under differential correlations. (new based on suggestions by Reviewer LgZi)

---

> ### Author Response · Authors · 2025-11-21
> **Q1: Neural results and discussion**
>
> Thank you for raising this interesting question. We provide predictions for both the tuning curves and the discrimination bound based on Table~1 in the manuscript. In this prediction, we assume $\alpha = 1$ (derived from calibration). Our model predicts the following:
>
> - The tuning-curve width scales on the order of $E^{-1}$.
> - The tuning-curve width is robust with respect to the neuronal noise variance $\eta_\kappa$.
> - The discrimination bound scales on the order of $E^{-3/2}$.
> - The discrimination bound scales on the order of $\sqrt{\eta_\kappa}$.
>
>
> Our prediction of the discrimination bound is derived from the Cramér–Rao bound, which can be attained by an unbiased efficient decoder. More specifically, under the assumption of a downstream unbiased and efficient neuronal decoder, the discrimination bound can be interpreted as the minimal stimulus difference that can be reliably discriminated at a given stimulus value.
>
> It is also worth noting that validating these predictions is challenging, as the effects of metabolic stress on single neurons and neuronal networks remain underexplored. However, we believe that the central motivation of this work – understanding how metabolic stress shapes neural coding – can be further advanced by future experimental investigations that directly probe metabolic constraints alongside neural response properties.

---

> ### Author Response · Authors · 2025-11-21
> **Q2: Additional validation of results in simulations**
>
> The simulations based on a stochastic Hodgkin-Huxley-like model aim to calibrate the parameters of $\alpha$ and $\eta_\kappa$. We extend the simulation for analyzing the coding effect and flattening effects with correlated noise among neurons in Sec. N. More specifically, we adopt differential correlations (Moreno-Bote et. al., Nature Neuroscience 2014), which has been shown to be the dominant factor in limiting information.
>
> During this simulation, we numerically solve the following relaxed optimization problem:
>
> $argmax_{g(\cdot)} \int p(s) \log \left(\frac{p(s)^2 g(s)^3 }{1 +  c p(s)^2 g^3(s) }\right) \mathrm{d}s - \lambda_{\text{penalty}} ||\int p(s) g(s) \mathrm{d}s - E||^2,$
>
> where $c$ is the differential correlation strength and the penalty hyperparameter $\lambda_{\text{penalty}}$ sets as $100$.
>
> As shown in the Fig. 11 (a newly added figure in revised manuscript), the optimal gain and density functions exhibit only modest changes across different levels of differential correlations; in contrast, metabolic stress produces a substantially larger effect (see Fig. 11a–b). We then derive the optimal tuning curves associated with the gain and density functions in Fig. 11a–b, which display the same flattening pattern predicted by our analytical solution in the infomax setting. In Fig. 11d, we quantitatively evaluate the relative broadening for each matched pair of correlation strengths in the control and food-restricted conditions. Although increasing the differential correlation strength from $0$ to $1000$ results in an additional 4% broadening, this effect is minor compared to the impact of metabolic stress, which produces approximately 38\% broadening. More implementation details are included in newly added Sec. N.
>
> This numerical experiment demonstrates that the flattening effect predicted by our proposed model remains robust even when differential correlations are taken into account.
>
> In summary, our simulation results (Secs. L2, L3, M, and N) highlight the following key properties of our model:
> - **(Secs. L2, L3)** The approximate homeostasis (AH) constraint is robust not only under Gaussian base tuning curves but also under Gabor bases. Moreover, this robustness holds across multiple coding objectives, including infomax, discrimax, and $\ell_1$ loss.
> - **(Sec. M)** Our proposed framework and analytical solution provide a substantially better account of the biological data reported in Padamsey et al.\ (2022) compared to competing approaches (Ganguli \& Simoncelli, 2010; Wang et al., 2016).
> - **(Sec. N)** The flattening effect predicted by our model remains robust even when differential correlations are incorporated.

---

### Author Response · Authors · 2025-12-02
**Global Rebuttal Response**

Before the summary, we note that Reviewer LgZi explicitly acknowledged that our revision generalizes the results to a more biologically relevant setting and effectively addresses the remaining clarity concerns. As a result, Reviewer LgZi updated their overall rating from 4 to 8, stating: "**I rise my score for Presentation to 3, my score for Contribution to 3, and the general Rating to 8**". Unfortunately, Reviewers iz34 and rMx9 did not provide comments during the rebuttal stage. Additionally, we combined the main paper and the Appendix into a single document to facilitate the discussion.

In the following summary, we denote Reviewers iz34, LgZi, and rMx9 as R1, R2, and R3, respectively. Additionally, Wx and Qx refer to the $x$-th weakness and question, respectively. We summarize the manuscript updates addressing the reviewers' comments as follows:
- **Model generalization (R1W3, R1Q2, R2W1, R3W1):** `Sec. N` now shows that our prediction of tuning-curve flattening under metabolic stress generalizes to differential correlations, the dominant factor limiting information transmission.
- **Model expressiveness and predictions (R1Q1, R3Q5):** `Lines 237-242` now explicitly state the model's prediction, providing a biophysically measurable objective for assessing neuronal properties under metabolic stress.
- **Recent works (R2W3, R2W3.5):** `Lines 140-147` emphasize the emergence of energy regularization in recent literature, highlighting new developments in energy-based analyses of neural networks.
- **Motivation (R2W4):** `Lines 44--51` clarify the paper's motivation and distinguish our model from related work.
- **Energy constraints and derivation of the objective (R2W2, R3Q2):** `Lines 57-64` clarify how prior studies model energy constraints. We further reiterate the novelty of our proposed constraints (Fig. 1 and Tab. 2) and clarify the derivation of the approximate objective in Sec. B.
- **Variable definitions (R3W2, R3W3, R3Q3, R3Q4, R3Q6):** `Lines 268-271 and 395-438`, along with Tabs. 3-4, clarify all variables. We additionally moved `Fig. 2` from the Appendix to the main text to illustrate how the simulation-related variables interact in the simulations for the calibration.
- **Simulation details (R2Q1, R2Q2):** `Lines 327-337` provide additional details of the simulation setup.
- **Link between data and discussion (R1W2):** `Lines 510-518` highlight how the model’s predictions connect to the data and how simulations calibrate model parameters, yielding accurate predictions of the tuning-curve flattening reported in Padamsey et al. (2022).
- **Approximate homeostasis constraint (R1W1, R3Q1):** We clarify that the approximate homeostasis (AH) constraint has not been formulated or applied previously in population-level coding. AH both generalizes earlier models (Fig. 1) and yields a transparent interpretation of $\alpha$; for example, $\alpha = 1$ and $\alpha = 1.5$ correspond to energy and coding-capacity constraints, respectively.

---

### Meta-Review · Area_Chair_9TFN · 2026-01-13

**Summary:**

The paper proposes a framework for how neural activity representations might adapt to reduce energy consumption during metabolic stress. The work extends optimal coding models by incorporating two novel constraints: approximate homeostasis and an ATP-based energy budget directly calibrated through biophysical simulations. The framework successfully predicts tuning curve flattening observed in food-restricted mouse visual cortex (Padamsey et al., 2022), which existing models fail to capture.

While initial reviews were mixed, all reviewers recommend acceptance following a rebuttal addressing concerns of biological realism and generalization.

**Reviewer Concerns:**

The reviewers expressed concerns of incremental technical novelty, limited numerical results and clarity and discussion. The authors address this criticism with new numerical experiments and comparisons to data. They also improved manuscript clarity and relationship of their model to experiment.

**Reviewer Scores:**

The one initially critical reviewer raised their score based on the rebuttal. The other two recommending weak accepts, would likely have kept the same scores.

---

### Decision · Program_Chairs · 2026-01-26

Accept (Poster)